



# Multi-million year cycles in modelled $\delta^{13}C$ as a response to astronomical forcing of organic matter fluxes.

Gaëlle Leloup[1,2] and Didier Paillard[2]

[1]Agence Nationale pour la gestion des déchets radioactifs (ANDRA), 1 Rue Jean Monnet, 92290 Châtenay-Malabry, France
[2]Laboratoire des Sciences du Climat et de l'Environnement, LSCE/IPSL, CEA-CNRS-UVSQ-Université Paris-Saclay, 91198 Gif-sur-Yvette, France

**Correspondence:** Gaëlle Leloup (gaelle.leloup@lsce.ipsl.fr)

**Abstract.**

Along with 400 kyr periodicities, multi-million year cycles have been found in $\delta^{13}C$ records over different time periods. A ∼8-9 Myr periodicity is found throughout the Cenozoic, and part of the Mesozoic. The robust presence of this periodicity in $\delta^{13}C$ records suggests an astronomical origin. However, this periodicity is barely visible in the astronomical forcing. Due

to the large fractionation factor of organic matter, its burial or oxidation produces large $\delta^{13}C$ variations for moderate carbon variations. Therefore, astronomical forcing of organic matter fluxes is a plausible candidate to explain the oscillations observed in the $\delta^{13}C$ records. So far, modelling studies forcing astronomically the organic matter burial have been able to produce 400 kyr and 2.4 Myr cycles in $\delta^{13}C$, but were not able to produce longer cycles, such as 8-9 Myr cycles. Here, we propose a mechanism that could explain the presence of multi-million year cycles in the $\delta^{13}C$ records, and their stability over time, as

a result of preferential periodicity locking to multiples of astronomical forcing periods. With a simple non linear conceptual model for the carbon cycle that has multiple equilibria, we are able to extract longer periods than with a simple linear model, and more specifically, multi-million year periods.

## 1 Introduction

Astronomical frequencies are imprinted into the carbon cycle. A 400 kyr oscillation has been seen in many $\delta^{13}C$ records

covering the Cenozoic : during the Paleocene (Westerhold et al., 2011), the Eocene (Lauretano et al., 2015), the Oligocene (Pälike et al., 2006), Miocene (Billups et al., 2004) and during more recent time periods (Wang et al., 2010). The 400 kyr frequency corresponds to one of the frequencies of eccentricity, and this is the most stable astronomical periodicity through geological times (Laskar, J. et al., 2004).

Longer cycles of 2.4 Myr have also been found in $\delta^{13}C$ records over the Cenozoic : over the Oligocene (Pälike et al., 2006;

Boulila et al., 2012), and Miocene (Liebrand et al., 2016). The 2.4 Myr periodicity is at the same time a true eccentricity cycle and present in the amplitude modulation (AM) of the 100 kyr and 400 kyr cycles (Laskar, J. et al., 2011; Boulila et al., 2012). ∼4.5 Myr cycles have also been observed in $\delta^{13}C$ in the late Creataceous (Sprovieri et al., 2013) as well as the latest Ordovician and Silurian (Sproson, 2020).

Much longer and dominant cycles of approximately 9 Myr have been found in $\delta^{13}C$ records, over the Cenozoic (Boulila et al.,



2012) and the Mesozoic (Martinez and Dera, 2015). Cycles of around 8 Myr have been seen in $\delta^{13}C$ in the late Creataceous (Sprovieri et al., 2013). This periodicity therefore seems very stable over time. The robust presence of $\sim$9 Myr cycles over various time periods, hints at an astronomical origin. However, in the case of 4.5 Myr and 9 Myr cycles, the link to the astronomical forcing is less clear, as these frequencies are not easily seen in the eccentricity spectra. A $\sim$4.5 Myr periodicity is visible in the eccentricity solution, although with a very low amplitude. The AM envelopes of the 100 kyr and 400 kyr

eccentricity also show a $\sim$4.5 Myr cyclicity, but again with low amplitude. The $\sim$9 Myr frequency is not directly visible in the eccentricity spectra. Although it has been suggested that the AM envelope of the $\sim$2.4 Myr filtered eccentricity has a $\sim$9 Myr cyclicity (Boulila et al., 2012), this remains with a very low amplitude. Therefore, the way in which the astronomy could pace these multi-million year cycles remains mysterious.

It has been suggested (Paillard, 2017; Kocken et al., 2019) that an astronomical forcing of organic carbon fluxes could explain

the observed $\delta^{13}C$ cyclicities at 400 kyr and 2.4 Myr. Indeed, organic, $^{12}C$ enriched, matter burial or oxidation can lead to relatively large $\delta^{13}$C variation for a moderate carbon variation, which is not possible with silicate weathering (Paillard, 2017; Russon et al., 2010). It is usually assumed that long term carbon cycle is mainly controlled by silicate weathering. But it is a negative feedback, that does not allow for oscillatory behaviour as observed in the data. Additionally, recent studies have highlighted the importance played by organic carbon fluxes in the long term carbon budget, acting as either a source if

petrogenic organic carbon is eroded and oxidized, or a sink if terrestrial organic carbon is exported and buried into sediments (Hilton and West, 2020). Organic carbon contributions are of the same order of magnitude than silicate weathering (Hilton and West, 2020). Therefore, astronomical forcing of the organic matter burial is a plausible candidate to explain the observed cyclic $\delta^{13}C$ variations.

Kocken et al. (2019) suggested that the link between $\delta^{13}C$ and the astronomical forcing could be explained by enhanced marine

organic carbon burial for lower eccentricity values. Periods of low eccentricity and therefore low seasonal contrast could favor annual wet conditions, and consequently clay formation. The majority of organic carbon burial is buried in association with clay particles (Hedges and Keil, 1995). If transport is not limiting, lower eccentricity would lead to higher marine organic carbon burial and increase $\delta^{13}C$. Paillard (2017) suggested a link between monsoons and organic carbon burial. Monsoon favor soil erosion and sediment transports. Recent soil as well as petrogenic organic carbon can be eroded and carried to

the oceans via rivers. The net effect on organic carbon burial depends on the geomorphological dynamics. For aggradational situations, that favour burial, the net result is organic matter burial. In that case, $\delta^{13}C$ increases with increased eccentricity and monsoon strength. However, for progradational situations, oxidation of petrogenic organic carbon is favoured, leading to a $\delta^{13}C$ decrease with eccentricity increase. Martinez and Dera (2015) suggested that low eccentricity values and the associated stable and humid conditions could favor constant freshwater and nutrient inputs, thus favoring water mass stratification, productivity

levels, leading to persistent anoxia and therefore higher organic carbon burial rates. On the contrary, high eccentricity values are associated with a recovery of oxic conditions during the cool season, limiting organic carbon burial. It has also been suggested that terrestrial organic carbon burial could increase with eccentricity minima, as this would favor annual soil anoxia (Kurtz et al., 2003). Laurin et al. (2015) suggested that $\delta^{13}C$ variations in the Late Cretaceous could be explained by a transient storage of organic matter or methane in quasi-stable reservoirs such as wetlands, soils, marginal zones of marine euxinic strata,



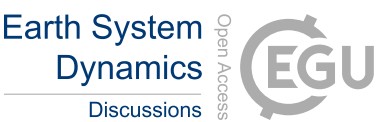

and permafrost. These quasi-stable reservoirs could respond non linearly to changes in obliquity and the consequent high latitude insolation changes or changes in meriodional insolation gradients. Other mechanisms have been proposed to explain $\delta^{13}C$ variation as a consequence from astronomical variations. de Boer et al. (2014) suggested that ice sheet dynamics and linked changes in carbon cycle dynamics could explain the 400 kyr cycles in $\delta^{13}C$. However, this mechanism does not permit to explain the presence of such cycles for periods without ice sheet (Kocken et al., 2019).

Modelling studies forcing the net organic matter burial through eccentricity have been able to produce 400 kyr and 2.4 Myr cycles in the $\delta^{13}C$ (Paillard, 2017; Kocken et al., 2019). However, these studies did not find longer cycles, such as the ~4.5 and ~9 Myr cycles observed in geological records. The model used were linear, and non linear mechanisms might be necessary in order to produce multi-million year cycles that have a very low amplitude in the input forcing.

Here, we develop a new conceptual model of the carbon cycle, that includes a net organic matter burial term and force it
astronomically with the eccentricity, similarly to the study of Paillard (2017). However, contrary to this study, the organic matter fluxes depend also on the surface carbon content, through its influence on climate, and on atmospheric oxygen content. Our model therefore couples the carbon and oxygen cycle, and is non linear, with the possibility of multiple equilibria. With this model, it is possible to obtain multi-million year frequencies in the $\delta^{13}C$. Depending on the strength of the astronomical forcing, periodicities of 2.4 Myr, ~4.8 Myr ~7 Myr, and ~9 Myr are produced preferentially, presumably as a result of
periodicity doubling of the 2.4 Myr eccentricity frequency.

## 2 Methods

### 2.1 Model description

Here, we develop a simple, yet non linear conceptual model, coupling the carbon and oxygen cycle. When forced by the eccentricity signal as input, the model is able to produce multi-million year frequencies as output, and multiples of the 2.4 Myr
frequency of eccentricity : 2.4 Myr cycles, ~4.5 Myr cycles, ~7 Myr cycles and ~9 Myr cycles.

Our model is based on the conceptual model of Paillard (2017). This model represents the evolution of the surface Earth carbon content, including the atmosphere, the ocean and the biosphere. This is opposed to carbon stored in deep soils, rocks or sediments. In our model, we have added a crude representation of the surface oxygen evolution. In the following, the features common to the Paillard (2017) model and ours are summarized and the differences are described.

In our model, the modeling of the surface carbon evolution is identical to the Paillard (2017) model. The surface carbon evolution is determined by the volcanic input V, the oceanic carbonate deposition flux D that is associated with silicate weathering, and the organic carbon burial B.

$$\frac{dC}{dt} = V - B - D \tag{1}$$

The organic matter burial B is meant as a net burial, and represents all organic carbon fluxes. $B = B^+ - B^-$ where $B^+$
represents organic carbon burial and $B^-$ represents organic matter oxidation.





As in Paillard (2017), we assume that the oceanic calcium concentration does not vary significantly over time and that carbonate compensation restores the oceanic carbonate content. With the use of an alkalinity balance (see Paillard (2017) for details), we obtain the following equation for the carbon evolution :

$$\frac{dC}{dt} = 2(V - B) - W \tag{2}$$

where W is the alkalinity flux to the ocean associated with silicate weathering. As in Paillard (2017), we assume silicate weathering to be the main stabilizer of the carbon system, with a fixed relaxation time $\tau_c$ : W = $C/\tau_c$. The volcanic input is considered as constant, $V = V_0$. The $\delta^{13}C$ evolution is described by :

$$\frac{d\delta^{13}C}{dt} = \frac{1}{C}(V(\delta^{13}V - \delta^{13}C) - B(\delta^{13}B - \delta^{13}C)) \tag{3}$$

We have assumed a constant -5‰ volcanic source ($\delta^{13}V$ = -5‰) and a constant -25‰ organic matter fractionnation ($\delta^{13}B - \delta^{13}C$ = -25‰). This slightly differs from the equation used in Paillard (2017), however the results remain very similar (for a detailed discussion, see the interactive discussion of the Paillard (2017) paper).

Compared to the previous model, we have added a crude representation of the oxygen cycle. Indeed, the oxygen interacts closely with organic matter burial and oxidation. On one side, the burial of organic matter is facilitated in low oxygen zones. On the other side, organic matter oxidation reduces the oxygen quantity, while burial of organic matter adds oxygen to the surface system. We consider a global oxygen content O, representing both the atmospheric $O_2$ and dissolved $O_2$ in oceans. On geological timescales, the atmospheric $O_2$ is driven by the net organic matter burial, the net burial of pyrite ($F_{pb} - F_{po}$ with $F_{pb}$ and $F_{po}$ the pyrite burial and oxidation), oxidation of volcanic gases ($F_v$) and oxidative weathering of sedimentary rocks, not already accounted in the net organic matter and pyrite burial, such as ferrous iron ($F_w$) (Berner, 2001; Canfield, 2005).

$$\frac{dO}{dt} = (B^+ - B^-) + (F_{pb} - F_{po}) - F_v - F_w \tag{4}$$

The oxidation of other elements than organic carbon is grouped in a single term, $Ox$, leading to :

$$\frac{dO}{dt} = B - Ox \tag{5}$$

We make the assumption that the oxidation of other elements than organic carbon increases linearly with oxygen contents : $Ox = a_{ox} \cdot O + b_{ox}$, with $a_{ox}$ and $b_{ox}$ two constants.

Although organic carbon perturbations are expected to have a weak impact on atmospheric $O_2$ concentrations, as atmospheric $O_2$ is relatively high in the Phanerozoic (Bergman et al., 2004; Berner and Canfield, 1989), perturbations to atmospheric $O_2$ of a few permill have been seen in Pleistocene ice cores (Stolper et al., 2016).

As in the Paillard model, the organic matter burial is forced astronomically, while the forcing term differs slightly in both models. Here, we choose to express the organic matter burial variations as :

$$B = B_0 - a_f F(t)$$

with F(t) = (e(t) -mean(e(t))) / max(e(t)) where e(t) is the eccentricity at time t, mean and max represent respectively the mean and maximum of eccentricity over the time period considered. $B_0$ represents the organic matter burial value without





astronomical forcing. The Analyseries software (Paillard et al., 1996) provides the La04 orbital parameters (Laskar, J. et al., 2004).

Several mechanisms have been proposed to be responsible of the link between organic matter burial and astronomical forcing.

Here, we do not intend to focus on a specific mechanism. We rather focus on the output signal that can be obtained from a simple astronomical forcing. Therefore, we have chosen the easiest possible relationship : a linear variation of the organic matter burial with the astronomical forcing. The negative sign was chosen in order to match the data, where eccentricity minima correspond to $\delta^{13}C$ minima, and thus organic matter burial maxima.

In the Paillard model, the $B_0$ term was constant and the organic matter burial therefore only varried with the astronomical

forcing. However, in our model, the organic matter burial depends not only on the astronomical forcing but also on the surface carbon content and the oxygen content, meaning $B_0 = B_0(C,O)$, as explained below.

Organic matter burial is facilitated in locally lower oxygen concentrations. All other things being equal, a higher oxygen content globally in the atmosphere will lead to higher oxygen contents locally in the ocean. Therefore, organic matter burial decreases for higher oxygen concentrations and inversely. Here, we have assumed an inverse linear dependency between the

organic matter burial and the oxygen content O. If $O_1$ and $O_2$ are two oxygen contents, then the difference in organic matter burial (for the same carbon content C) is :

$$B_0(C,O_1) = B_0(C,O_2) - \delta(O_1 - O_2) \tag{6}$$

$\delta$ is a positive constant. For a similar carbon content C, the organic matter burial is higher in the case of lower oxygen quantities.

Here, we also suggest that net organic carbon burial depends on the surface carbon quantity C. Indeed, climate can influence the organic matter burial, and as a first approximation, larger carbon values C in the surface system correspond to warmer, wetter climates. Net organic matter burial contribution to the surface carbon evolution is composed of two opposite contributions : biospheric organic carbon erosion leading to $CO_2$ drawdown ($CO_2$ sink) and therefore surface carbon decrease, and oxidation of petrogenic organic carbon, leading to $CO_2$ rise ($CO_2$ source), and therefore surface carbon increase. Eroded terrestrial

organic matter from plants is delivered to rivers (Meybeck, 1982; Ludwig et al., 1996). If a part of this biospheric organic carbon is buried into sediments without being degraded, this corresponds to a decrease of the surface carbon content. It has been estimated that the current burial flux of organic carbon eroded from land into oceanic sediments is around 40-80 MtC/yr (Hilton and West, 2020). On the other hand, exhumation of sedimentary rocks can lead to the oxidation of petrogenic organic carbon and therefore to $CO_2$ release (Hilton et al., 2014). The carbon flux released to the atmosphere through petrogenic or-

ganic carbon oxidation has been estimated to be between 40 and 100 MtC/yr (Hilton and West, 2020). Climate can act on these processes on several ways. When the surface carbon content increases, this globally results in a warmer and wetter climate. Warmer temperatures and stronger runoff increase erosion and transfer of biospheric organic carbon (Hilton, 2017; Smith et al., 2013). In addition, warmer temperatures increase ocean stratification and decrease the solubility of oxygen in surface waters (Bopp et al., 2002), leading to expansion of oxygen minimum zones (Stramma et al., 2008, 2010). This decreases organic

matter oxidation and favours its burial into oceanic sediments (Jessen et al., 2017). Other climatic related factors, have been





suggested to limit dissolved oxygen in the ocean, such as increased phosphorus inputs (Baroni et al., 2020; Niemeyer et al., 2017). These inputs are expected to increase for warmer and wetter climate, that increases weathering, leading to regional deoxygenation and organic carbon burial (Baroni et al., 2020).

It has also been suggested that the oxidation of petrogenic organic carbon could be linked to climate as petrogenic organic carbon oxidation could be locally limited by temperature, $O_2$ contents, and microbial activity (Chang and Berner, 1999; Bolton et al., 2006; Hemingway et al., 2018; Petsch et al., 2005). Higher temperature could lead to stronger petrogenic carbon oxidation. We have made the assumption that for intermediate carbon value, and thus intermediate temperatures, this mechanism is dominant, whereas the increase in export of biospheric carbon and increased burial dominates for warmer temperatures (larger C values). We have supposed that there is no dependency to climate for lower carbon values. This results in a non monotonic dependency of B to C. For low carbon values ($C < C_1$) and thus colder climates, the organic carbon burial B does not depend on C. Then, for intermediate carbon values ($C_1 < C < C_2$), the organic carbon burial decreases with increasing temperatures, and thus increasing carbon C. Finally, for higher carbon values ($C > C_2$) the organic carbon burial increases with increasing temperature (and thus, carbon C). For the sake of simplicity, we have made the assumption of linear variations. The resulting shape of the evolution of organic carbon burial fluxes as function of the carbon content C is represented in Figure 1.

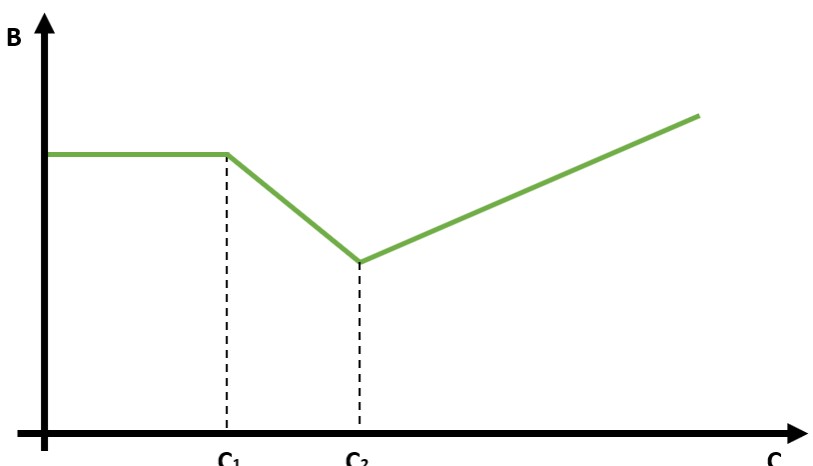

**Figure 1.** Schematic representation of the organic carbon burial B evolution with carbon content C

These assumptions on the dependency of the organic matter burial flux to the surface carbon content C are strong. It is not an easy task to quantify at present day the net magnitude of carbon burial, as some regions are known to be carbon sinks and others known to be carbon sources. It is an even more complicated task to have an estimation of this budget for different climates, and therefore different carbon contents. However, the results discussed below do not depend on the specific shape chosen for the dependency of the organic matter burial B to the carbon content C. Similar results could be obtained with an other dependency, as long as it is non monotonic with the carbon content C and we place ourselves here in one of the simplest case possible. The shape of the organic burial evolution is certainly different and more complex than the one presented here. However, as many





process act on the net organic matter burial, some favoring organic matter burial, and some favoring organic matter oxidation, we are confident in the fact that the relationship between organic matter burial and surface carbon content is non monotonic. In our model, carbon and oxygen evolution depend on the organic matter burial term, that conversely depends on both, carbon and oxygen quantities. Therefore, in our model the carbon and oxygen cycle are coupled via the organic matter burial term. This is the main difference with the model of Paillard 2017, as this results in a non linear model, where it is possible to have multiple equilibria in the carbon and oxygen system, and thus oscillations even without astronomical forcing.

## 2.2 Parameter values

We first consider the case where the organic matter burial is not forced astronomically ($a_f = 0$). For simplicity, we have chosen $B_0$ to be piecewise constant when considering the variations due to carbon. It is decreasing linearly with oxygen. Therefore, we can write $B_0$ as :

$$B_0(C,O) = B_0(C,O_{ref}) - \delta(O - O_{ref}) \tag{7}$$

with

$$B_0(C,O_{ref}) = \begin{cases} \alpha & \text{for C} < C_1 \\ \alpha - \beta(C - C_1) & \text{for } C_1 < \text{C} < C_2 \\ \alpha - \beta(C_2 - C_1) + \gamma(C_2 - C_1) & \text{for C} > C_2 \end{cases} \tag{8}$$

Where $\delta$ ($\delta > 0$) is the coefficient representing the strength of the organic matter burial evolution with oxygen. The higher $\delta$, the stronger is the organic matter burial decrease with oxygen content. $\alpha$ represents the constant value of organic matter burial for low carbon values ($C < C_1$). $-\beta$ ($-\beta < 0$) represents the slope of the carbon burial evolution for intermediate carbon values ($C_1 < C < C_2$), for which the organic matter burial decreases with increasing carbon content. $\gamma$ ($\gamma > 0$) represents the slope of the carbon burial evolution for high carbon values ($C > C_2$), for which the organic matter burial increases with increasing carbon content. We choose to place ourselves in the case where there are two stable equilibria for the carbon cycle for the current oxygen value (see section 3.1). This corresponds to panel (a) of Figure 2. These equilibria are called $C_{eq1ref}$ and $C_{eq2ref}$.

Our unforced model therefore contains 13 parameters : the value of carbon emissions associated with volcanism $V_0$, the time constant associated with silicate weathering $\tau_c$, the current oxygen content $O_{ref}$, the two values of the equilibria for $O = O_{ref}$, $C_{eq1ref}$ and $C_{eq2ref}$, parameters linked to the shape of the evolution of organic carbon burial with carbon ($C_1$, $C_2$, $\beta$, $\gamma$, $\alpha$), one parameter linked to the evolution of organic matter burial with oxygen ($\delta$), and two parameters linked to the evolution of oxidation of other elements than carbon with oxygen ($a_{ox}$ and $b_{ox}$). When the organic carbon burial is forced astronomically, there is one additional parameter, representing the strength of the astronomical forcing : $a_f$.

$O_{ref}$ is the current oxygen level and is equal to $O_{ref} = 1.19 \cdot 10^6$ Pg. $C_{eq1ref}$ and $C_{eq2ref}$ are the stable equilibria for $O = O_{ref}$. We have choosen $C_{eq1ref}$ to be equal to the pre industrial carbon content of Earth's surface reservoirs (atmosphere, oceans and biosphere). It is estimated to be around 43 000 PgC (Ciais et al., 2013). We have chosen the second equilibria to be





at higher carbon values, $C_{eq2ref}$ = 47000 PgC. The $\tau_c$ constant is set to 200 kyr (Archer et al., 1997). The $V_0$ value was chosen in order to have $\delta^{13}C$ values around zero for medium carbon values, $V_0 = (5/8) \cdot (1/2)(C_{eq1ref} + C_{eq2ref})/\tau_c = 140\,\mathrm{TgCyr}^{-1}$. This is a value slightly higher than in Paillard (2017). However, it remains in the range of possible values for carbon emissions

from volcanism, estimated between 40 and 175 TgC/yr (Burton et al., 2013). That $C_{eq1ref}$ and $C_{eq2ref}$ are equilibria for $O = O_{ref}$ in the unforced case gives constraints on the $\alpha$ and $\gamma$ parameters. Several values of $\beta$ remain possible, with the constraint that we should have $\beta > 1/2\tau_c$. $C_1$ and $C_2$ should be contained between $C_{eq1ref}$ and $C_{eq2ref}$ : $C_{eq1ref} < C_1 < C_2 < C_{eq2ref}$. In the following, we have taken $\beta = 2.4 \cdot 1/2\tau_c$, $C_1 = C_{eq1ref} + (1/3)(C_{eq2ref} - C_{eq1ref})$, $C_2 = C_{eq1ref} + (2/3)(C_{eq2ref} - C_{eq1ref})$. Changing these parameters would change slighlty the form of the unforced oscillations. The $\delta$ parameter influences

the size of the free oscillations for the oxygen. We have set $\delta$ in order to have free oscillations of the size of a few percent of the reference oxygen value. This corresponds to $\delta \sim 2 \cdot 10^{-7}$. Different values of the parameters $a_{ox}$ and $b_{ox}$ are studied in the following, corresponding to the four possible different cases outlined by the four panels of Figure 3 (see Section 3.1).

## 3 Results and discussion

Here we study the results for three different cases to examine the importance of the different processes.

  – In a first step, we do not consider astronomical variations of the organic matter burial B. Therefore, organic matter burial only depends on the carbon and oxygen quantities, $B = B_0(C,O)$.

  – In a second step, we consider a formulation similar to the model of Paillard (2017), where the organic matter burial depends on the astronomical forcing but does not depend on the carbon and oxygen quantities : $B = B_{0cst} - a_f F(t)$ with $B_{0cst}$ being a constant.

  – In a last step, we consider that the organic matter burial depends on both the astronomical forcing and the carbon and oxygen contents (complete form) : $B = B_0(C,O) - a_f F(t)$

### 3.1 Results without astronomical forcing of the organic matter burial

First, we look at our model results in the case where the organic matter burial is not forced astronomically, corresponding to $a_f = 0 : B = B_0(C(t),O(t))$. This is done in order to better understand the dynamics of the coupled oxygen/ carbon system

without astronomical forcing. We first consider the system from a theoretical point of view, by describing the different possible equilibria. Then, we compute the results numerically for different parameter values.

Equilibria for the carbon are defined as $\frac{dC}{dt} = 0$. From equation 2, we get :

$$\frac{1}{2}\frac{dC}{dt} = (V - \frac{W}{2}) - B \tag{9}$$

Carbon equilibria are thus equivalent to $B = V - W/2$. As we have assumed $W = C/\tau_c$, and V to be constant, $V = V_0$, this

corresponds to $B = V_0 - C/(2\tau_c)$. Figure 2 represents schematically the evolution the organic matter burial term and the V - W/2 term as a function of the carbon content C. Depending on the relative position of the B curve to the V - W/2 curve, this





leads to one or two stable equilibria. For clarity, in the following we will call the B curve (green in the Figures) the organic term, and the V - W/2 curve (red curve) the inorganic term.

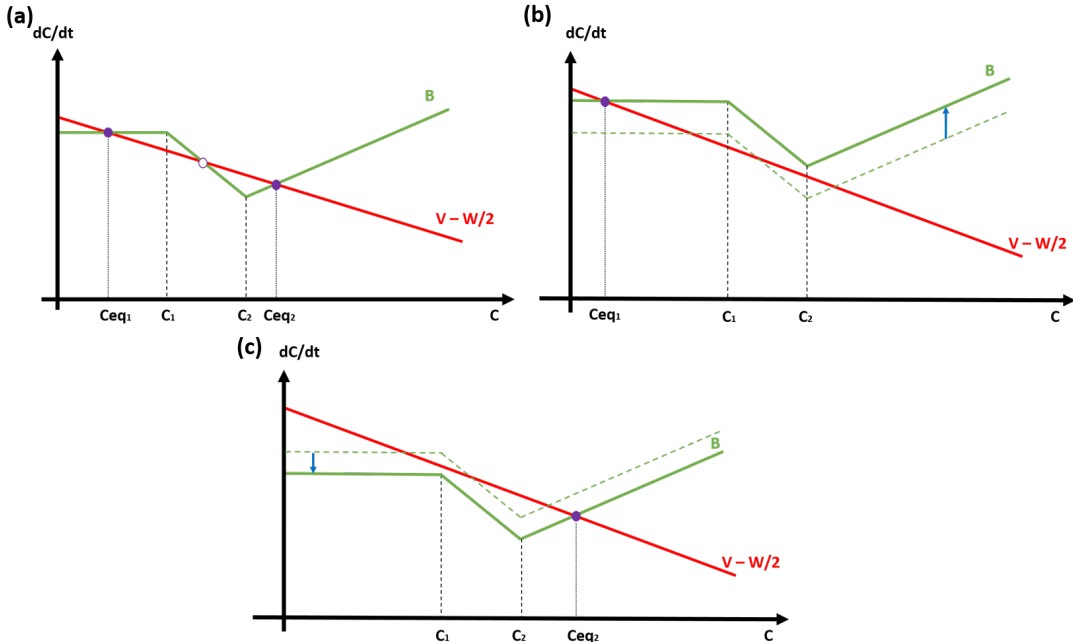

**Figure 2.** Schematic representation of the organic (green, B) and inorganic (red, $V_0 - C/(2\tau_c)$) terms as a function of the carbon content C. (a) Case of two equilibria in the carbon system ($C_{eq1}$ and $C_{eq2}$). (b) Case of one equilibria in the carbon system, at low carbon value ($C_{eq1}$). (c) Case of one equilibria in the carbon system, at high carbon value $C_{eq2}$.

In the case represented by panel (a) of Figure 2, the organic and inorganic curves cross at three different locations, meaning

that there are three different carbon values C for which an equilibrium is obtained ($dC/dt = 0$). However, only two of these equilibria are stable and are indicated by full purple circle, while the unstable equilibria is indicated by an empty purple circle. In the case of the empty circle, if we diverge a little from the equilibria value to higher carbon values, then V - W/2 > B (the red curve is above the green one) and thus dC/dt >0. This small divergence to higher carbon value, leads to even higher carbon values : the equilibrium is unstable. The same reasoning can be done if we diverge a little from the equilibrium towards lower

carbon values. On the contrary, if we consider the first full purple point from panel (a) of Figure 2, it is a stable equilibrium. If we diverge a little from this equilibrium towards higher carbon values, then B > V - W/2 (the green curve is above the red) and thus dC/dt < 0. The system is brought back towards lower carbon values : the equilibrium is stable. Panel (a) of Figure 2 represents the case of two stable equilibria in the carbon system. However, it is possible to have only one stable equilibrium in the carbon system, if the value of B is higher or lower, as is shown in panels (b) and (c) of Figure 2. In the

case of a higher organic matter burial, there is only one crossing between the green and red curves as represented in panel (b) of Figure 2 : there is only one equilibrium for the carbon cycle, $C_{eq1}$. A similar reasoning than previously shows that this





equilibrium is stable. In a similar manner, in the case of a lower organic matter burial, there is only one crossing between the green and red curves as represented in panel (c) of Figure 2 : there is only one equilibrium for the carbon cycle, $C_{eq2}$. This equilibrium is stable. These three configuration of the organic matter burial correspond to different oxygen values. Indeed, as

$B(C,O) = B(C,O_{ref}) - \delta(O - O_{ref})$, higher oxygen values correspond to a shift of the organic matter (green) curve towards the bottom, and conversely. Therefore, it is possible to switch between the configurations, if the oxygen evolves.

Similarly to what has been done for the carbon cycle, we look at possible equilibria in the oxygen cycle. In our model, dO/dt = 0 corresponds to $B = Ox$, meaning that the organic matter burial is equal to the oxidation of other elements than carbon. These terms are represented in Figure 3. The four different configurations shown in panels (a) - (d) correspond to different values for

the slope and intercept of the $Ox$ curve. For these four configurations, the shape of the organic matter burial B (green) curve is similar. For low oxygen values ($O < O_{lim2}$), there is only one possible value of the organic matter burial B for each oxygen value. This corresponds to the case displayed in panel (b) of Figure 2, where there is only one possible equilibrium value for the carbon content C, and therefore only one possible value of B for a given oxygen value. For decreasing oxygen value, the organic burial increases. For high oxygen values, ($O > O_{lim1}$), there is also only one possible value of the organic matter burial

B for each oxygen value. This corresponds to the case displayed in panel (c) of Figure 2. There is only one possible value for the carbon content C, and therefore only one possible value of B for a given oxygen value. These values of organic carbon burial B are lower than the one obtained with $O < O_{lim2}$. For intermediate oxygen values ($O_{lim2} < O < O_{lim1}$) there are two possible equilibrium values for the carbon content C, and therefore two possible equilibrium values for the organic matter burial B for a given oxygen value. This corresponds to the case displayed in panel (a) of Figure 2. The unstable equilibrium is

displayed with dotted line.

Following our hypothesis, the $Ox$ curve is simply a straight line. However, depending on the slope and intercept of the $Ox$ curve, this leads to four different configurations, where the (blue) $Ox$ curve either crosses the upper branch of B, the lower branch of B, none of them or both of them. These four different possibilities are schematized in Figure 3. When the $Ox$ curve crosses the upper branch of B (Panel (b) of Figure 3), there is one possible equilibrium value for the oxygen (dO/dt = 0),

represented by the full orange circle. It is a stable equilibrium : if we divert to lower oxygen values, then B > Ox (the green curve is above the blue line), meaning that dO/dt > O, and the oxygen value increases back towards equilibrium. Similarly, when the $Ox$ curve crosses the lower branch of B (Panel (c) of Figure 3), there is one possible equilibrium value for the oxygen. This is also a stable equilibrium. When the $Ox$ curve crosses both the upper and lower branches of B (Panel (d) of Figure 3), it crosses also the middle, unstable, branch. There are three possible equilibrium values for the oxygen, two of which are stable

(the crossing with the upper and lower branch). For these three cases (panels (b), (c) and (d) of Figure 3), the system will converge towards an equilibrium value ($C_{eq}$, $O_{eq}$) and remain in this state as this are stable equilibria. When the equilibrium is reached the carbon and oxygen content will not evolve anymore. The organic matter burial and the $\delta^{13}C$ will therefore also not change.

However, the situation is different in the case where the $Ox$ curve does not cross the lower or upper branch of B, but crosses

the middle branch (dashed line), as represented in Panel (a) of Figure 3. This branch is not stable from the carbon point of view (it corresponds to the unstable equilibria of Figure 2). It is also not a stable equilibria for oxygen. In this case, no equilibrium is



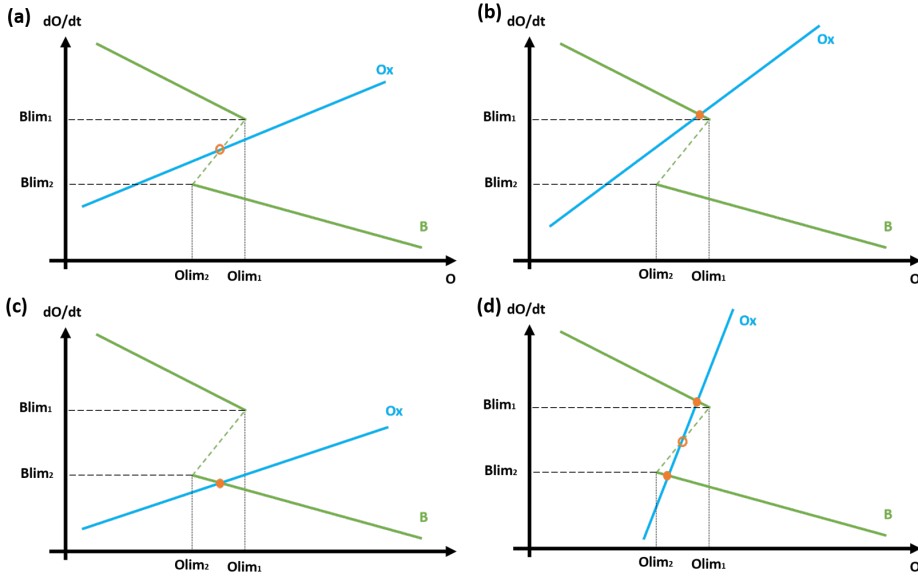

**Figure 3.** Schematic representation of the organic term (B, green curve) and the Ox term (blue curve) as a function of the oxygen content O. Four different cases are displayed, depending on the slope and intercept of the $Ox$ curve : (a) one unstable equilibrium; (b) and (c) a stable equilibria; and (d) two stable equilibria

reached for the (C, O) system, and oscillations of the oxygen and carbon content can be obtained without astronomical forcing of the organic matter burial. The organic matter burial and the $\delta^{13}C$ therefore also oscillate.

To illustrate these cases, we run our model without astronomical forcing. For conciseness reasons, we consider only two out

of the four possible cases described above. We will consider the two cases for which the $Ox$ curve crosses the unstable middle branch, corresponding to panels (a) and (d) of Figure 3. We define the parameter $a_{lim}$ as $a_{lim} = (B_{lim1} - B_{lim2})/(O_{lim1} - O_{lim2})$. If $a_{ox} > a_{lim}$, this corresponds to the case of panel (d) of Figure 3. If $a_{ox} < a_{lim}$, this corresponds to the case depicted in panel (a) of Figure 3.

### 3.1.1  $a_{ox} > a_{lim}$

The model is run for 100 Myr with $a_{ox} = 1.5 \cdot a_{lim}$, for one specific set of model parameters. The simulations start from different initial values for the carbon and oxygen. The $Ox$ curve intercept parameter $b_{ox}$, is chosen in order to follow the configuration of panel (d) of Figure 3, where the $Ox$ curve crosses 3 times the B curve. The carbon, oxygen, organic matter burial and $\delta^{13}C$ evolution over time are displayed in Figure 4. An equilibrium is reached in less than 10 Myr for all the simulations. The equilibrium value depends on the initial conditions. This is expected as there are two stable equilibria for the

(C,O) system in this configuration. In the case of a different value of the parameter $b_{ox}$, leading to a single crossing with the B curve (case of panel (b) and (c) of Figure 3), an equilibrium would also be reached. The only difference is that the equilibrium would not depend on initial conditions, as there is only one possible equilibrium value.



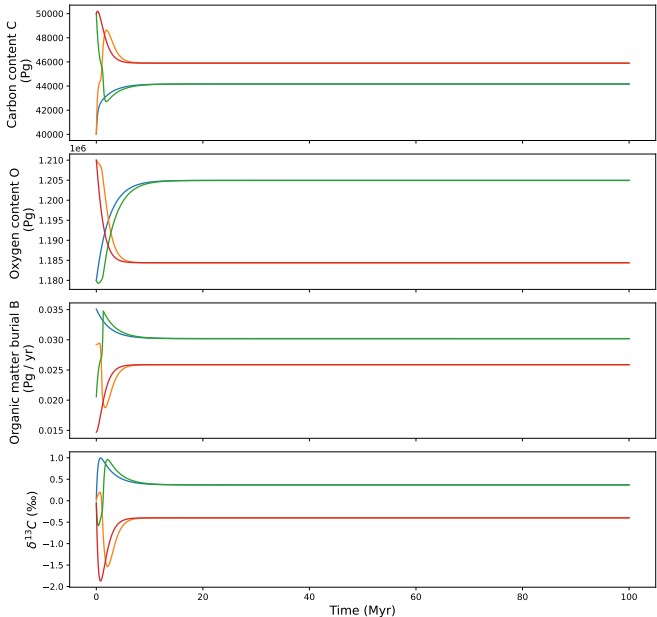

**Figure 4.** Modelled carbon content C, oxygen content O, organic matter burial B and $\delta^{13}C$ evolution in the case without astronomical forcing for different initial states. Case $a > a_{lim}$.

### 3.1.2 $a_{ox} < a_{lim}$

The model is run for 100 Myr with $a_{ox} = 0.5 \cdot a_{lim}$, for one specific set of model parameters, starting from different initial values for the carbon and oxygen. The parameter $b_{ox}$, is chosen in order to follow the configuration of panel (a) of Figure 3, where the $Ox$ curve only crosses the unstable part of the B curve. The carbon, oxygen, organic matter burial and $\delta^{13}C$ evolution over time are displayed in Figure 5.

In this case, no equilibrium is reached and the system oscillates. The amplitude of the oscillations for the carbon is around 6000 PgC. The $\delta^{13}C$ oscillations have an amplitude around 2.5‰. The frequency of the oscillation is identical for the different initial conditions, and only the phase is different. The oscillations have a period of approximately 15 Myr with the parameter set used on this example. The frequency and shape of the oscillations depend on the model parameters. For example, for higher values of the $\delta$ parameter, the system would oscillate more quickly.

To sum up, we have detailed here a simple case, without astronomical forcing of the organic matter burial B. In that case, there are two possibilities for the evolution of the (C, O) system. If there is at least one stable equilibrium for the oxygen (case of panels (b), (c) and (d) of Figure 3), then the system will converge towards an equilibrium $(C_{eq}, O_{eq})$, and the modelled $\delta^{13}C$ will also converge towards an equilibrium. If there is no stable equilibrium for the oxygen (case of panel (a) of Figure 3), the

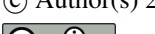



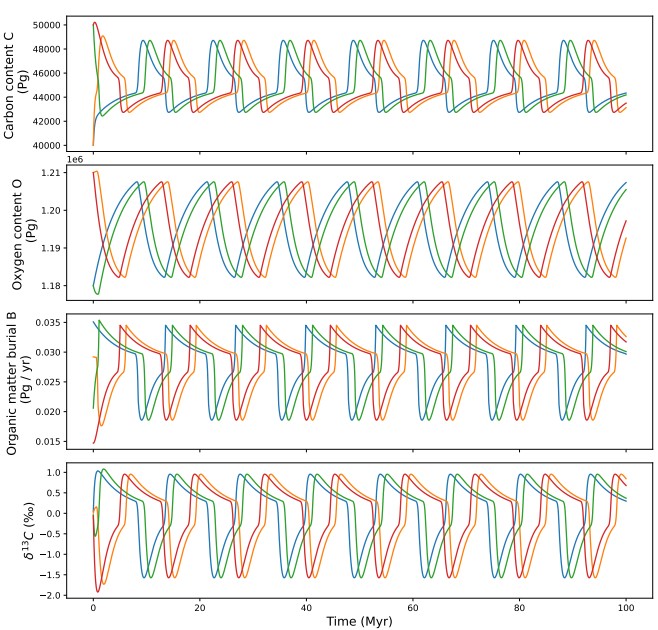

**Figure 5.** Modelled carbon content C, oxygen content O, organic matter burial B and $\delta^{13}C$ evolution in the case without astronomical forcing for different initial states. Case $a < a_{lim}$.

(C,O) system will oscillate freely, and the modelled $\delta^{13}C$ will also oscillate. However, in our model, the organic matter burial is also forced astronomically, through a dependency to eccentricity. This makes the situation more complex. When a stable equilibrium is reached in a case without astronomical forcing, the equilibrium can become unstable in the forced case, if the astronomical forcing is relatively strong (relatively high values of the $a_f$ parameter). In this case, the astronomical forcing pushes the system away from its equilibrium towards another. This will be detailed in the third result section.

### 3.2 Results without oscillatory dynamics - organic matter burial not depending on surface oxygen.

In a second step, we study our model results in the case where the net organic matter burial does not depend on surface carbon and oxygen quantities, but is solely forced by the astronomical forcing : $B = B_{0cst} - a_f F(t)$, where $B_{0cst}$ is a constant. This is close to the model of Paillard (2017). In this case we have no coupling anymore between the carbon and the oxygen cycle, and the system is linear, with an astronomical forcing.

$$\begin{cases} \frac{dC}{dt} = -\frac{1}{\tau_c}C + c_1 - 2a_f F(t) \\ \frac{dO}{dt} = -a_{ox}O + c_2 + a_f F(t) \end{cases} \tag{10}$$





with $c_1$ and $c_2$ being constants, $c_1 = 2(V_0 - B_0)$, $c_2 = B_0 - b$. As $a_{ox} > 0$ and $\frac{1}{\tau_c} > 0$ the system is stable. Without the astro-
nomical forcing, it would converge towards an equilibrium. The addition of the astronomical forcing yields oscillations around

it.

In the following, we have run the model with the same parameters as previously, for different values of the $a_f$ parameter. In
Figure 6 the evolution of carbon, oxygen, organic carbon burial and $\delta^{13}C$ over time are displayed for $a_f = 0.01$ (green curves),
$a_f = 0.03$ (orange curves) and $a_f = 0.05$ (blue curves). There are oscillations around an equilibrium value for the carbon and the
$\delta^{13}C$. Changing the $a_f$ value does not change the shape nor the frequency of the oscillations. It only changes their amplitude.

For the $\delta^{13}C$, the oscillation amplitude varies from around $0.5‰$ for the weakest astronomical forcing considered ($a_f = 0.01$)
to $2‰$ for the strongest astronomical forcing considered ($a_f = 0.05$). For the carbon, the oscillation amplitude varies from
around 1000 PgC to 6000 PgC.

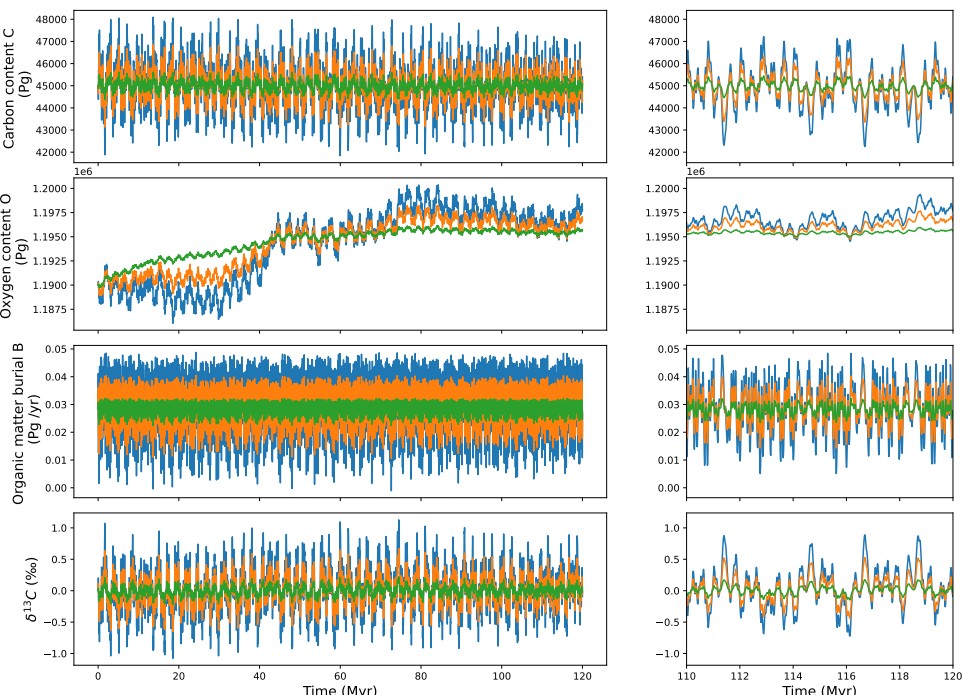

**Figure 6.** Modelled carbon content C, oxygen content O, organic matter burial B and $\delta^{13}C$ for different $a_f$ values ($a_f = 0.01$ in green, $a_f = 0.03$ in orange and $a_f = 0.05$ in blue) when $B_0$ is a constant. The right panel is a zoom on the last 10 Myr of the simulation.

The spectral analysis of the $\delta^{13}C$ curve is shown in panel (a) of Figure 7. The dominant frequencies of the $\delta^{13}C$ oscillations
are 400 kyr and 2.4 Myr. These frequencies are present in the input forcing (the eccentricity). The 400 kyr frequency is already


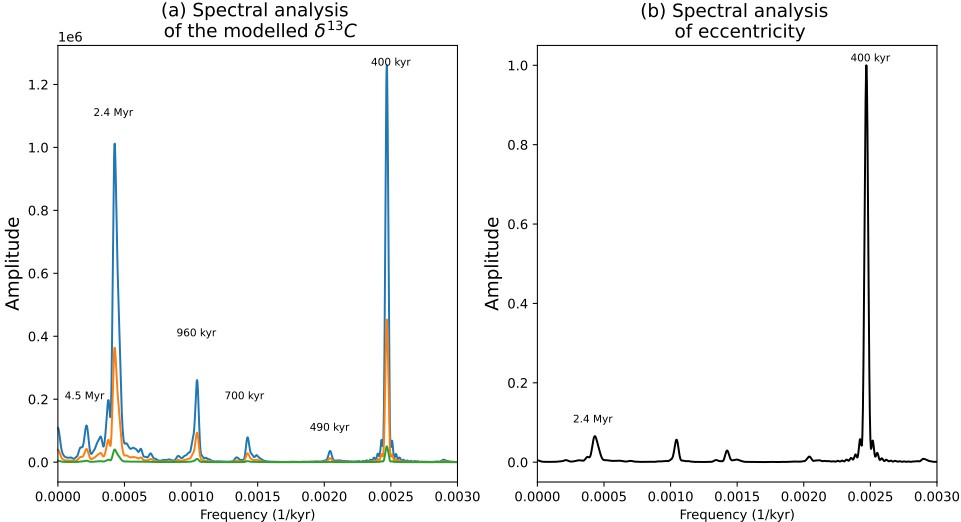

**Figure 7.** (a) Normalized spectral analysis of the resulting $\delta^{13}C$ for different $a_f$ values ($a_f$ = 0.01, 0.03 and 0.05 in green, orange and blue) when $B_0$ is a constant. (b) Spectral analysis of the eccentricity input forcing.

strong in the input forcing. However, the 2.4 Myr frequency is of low power in the eccentricity spectra, but strong in the modelled $\delta^{13}C$. The $\delta^{13}C$ spectral analysis also contains a weaker $\sim$4.5 Myr peak. This frequency is of very low amplitude in the input eccentricity forcing. As in the Paillard (2017) model, we are able to produce 400 kyr and 2.4 Myr cycles in $\delta^{13}C$ through the astronomical forcing of organic matter burial. However, as this model is linear, it is not possible to produce longer term cycles with a high amplitude, corresponding to frequencies with very low amplitude in the eccentricity spectra. A non

linear model is needed to produce larger periods with a strong amplitude in $\delta^{13}C$ with a simple eccentricity forcing. This is the case with our model when we add the dependency of the organic matter burial to the carbon and oxygen contents. The results for specific parameter values are discussed in the next section.

### 3.3   Complete model : forcing of the organic matter burial.

Here, we use the complete form of the organic matter burial, as described in Section 2.1. The organic matter burial depends on

the surface carbon quantity C, the surface oxygen quantity O and the astronomical forcing : $B = B_0(C(t), O(t)) - a_f F(t)$. We perform simulations for different $a_f$ parameter values ranging from $a_f$ = 0 to $a_f$ = 0.05. This corresponds to different relative importance of the astronomical forcing on the organic matter burial. The simulations are run over 120 Myr. We perform these simulations in two different cases. The first [case A] corresponds to a situation where the system oscillates freely for $a_f$ = 0. This corresponds to panel (a) of Figure 3. The second one [case B] corresponds to a situation where the system does not

oscillate for $a_f$ = 0 (presence of a stable equilibria). In our case, we have taken a situation corresponding to panel (b) of Figure 3. The evolution of $\delta^{13}C$ in these both cases is displayed in panels (a) of Figures 8 and 9. A zoom on the last 20 Myr of





simulation is provided in panels (b). The spectral analysis of the modelled $\delta^{13}C$ is displayed in panels (c). The spectral analysis were carried out with the Blackman Tuckey method implemented in the Analyseries software (Paillard et al., 1996).

In case A, the $\delta^{13}C$ oscillates freely when the system is not forced astronomically ($a_f = 0$), with an amplitude around 2‰.

For a relatively small influence of the astronomical forcing ($a_f = 0.01$) the shape of the large, free oscillations is still visible, and small oscillations of amplitude around 0.1‰ occur around it. However, the frequency of the free oscillations is affected by the astronomical forcing and differs from the unforced case. In case B, there are no oscillations for $a_f = 0$. For $a_f = 0.1$, oscillations similar to case A occur. In both cases, for higher values of $a_f$, the free oscillations are not visible anymore, and the signal is dominated by oscillations of lower period. The dominant, low period oscillations have an amplitude of around

2‰, and the smaller higher frequency oscillations have an amplitude of around 0.05‰. Oscillations of 400 kyr are present for all cases where the system is forced astronomically. However, their relatively low amplitude makes them hard to notice in the spectral analysis, especially for low $a_f$ value. As the relative strength of the astronomical forcing increases ($a_f$ increases), the power of the 400 kyr oscillations becomes stronger. The dominant period of the oscillations decreases with increasing $a_f$ parameter.

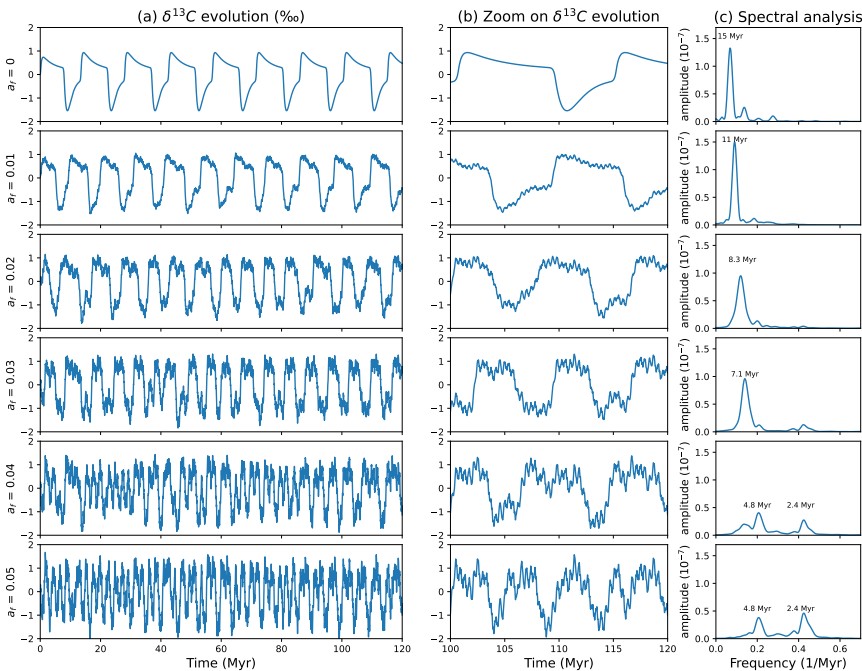

**Figure 8.** [case A] (a) Modelled $\delta^{13}C$ for different $a_f$ values and $a = 0.5 \cdot a_{lim}$. (b) Zoom on the last 20 Myr of the simulation. (c) Spectral analysis of the modelled $\delta^{13}C$.



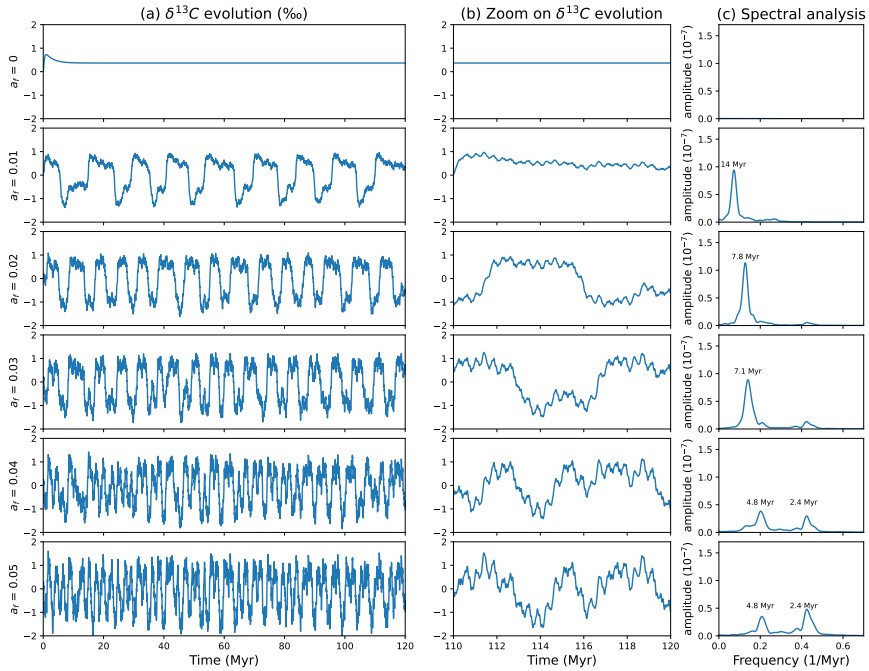

**Figure 9.** [case B] (a) Modelled $\delta^{13}C$ for different $a_f$ values and $a = 1.5 \cdot a_{lim}$. (b) Zoom on the last 20 Myr of the simulation. (c) Spectral analysis of the modelled $\delta^{13}C$.

In both cases, the addition of the astronomical forcing changes the behaviour of the coupled system. The output signal has a dominant frequency that differs from the frequency of the unforced system. We are able to obtain oscillations on the $\delta^{13}C$ of several million years. However, one important feature to notice is that when the $a_f$ parameter decreases, the output signal dominant period does not decrease continuously. Figure 10 represents the dominant period of the signal depending on the $a_f$ parameter, for different parameter sets. The plot shows "steps" : there are preferential periods, meaning that for a range range

of $a_f$ values, the output dominant period remains the same. The 2.4 Myr periodicity and its multiples are schematized with blue dotted lines.

    For large $a_f$ values (around 0.4 - 0.5), the dominant frequency is 2.4 Myr. Both 2.4 and 4.8 Myr periodicities are present in the spectral analysis. As the $a_f$ parameter decreases slightly, the 4.8 Myr periodicity becomes dominant. It is particularly striking that for each parameter set, it is not possible to have a dominant frequency value between 2.4 and 4.8 Myr. The 4.8

Myr periodicity is the double of the 2.4 Myr periodicity of eccentricity. As the $a_f$ parameter decreases, a period around 7 Myr becomes dominant. It is between 7.2 and 7.7 depending on the parameter set. This periodicity is not present in the eccentricity spectra. However, it corresponds to the triple of the 2.4 Myr periodicity. It is possible to obtain dominant frequencies between





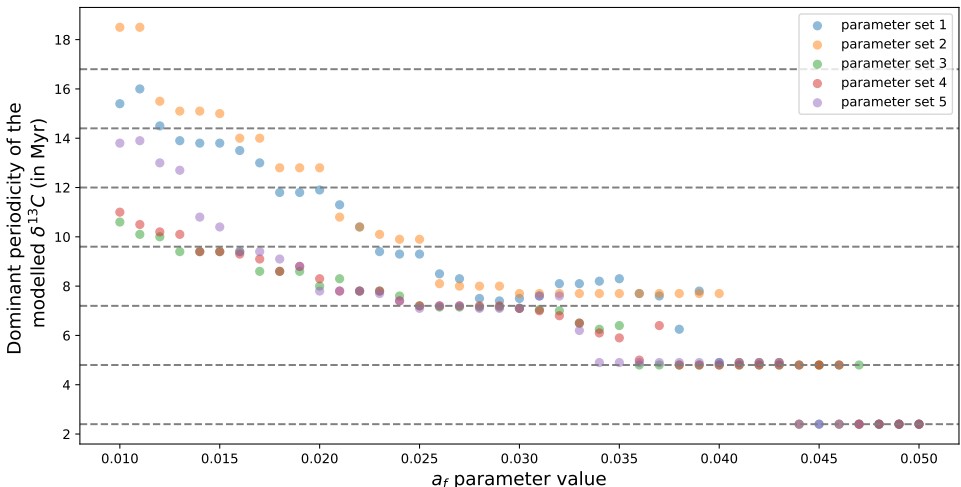

**Figure 10.** Dominant period of the modelled $\delta^{13}C$ as a function of the $a_f$ value, for different parameter sets.

4.8 and 7 Myr. However, this is limited to very specific cases (specific $a_f$ values for specific parameter values). On the contrary, the $\sim$7 Myr period is dominant for a wide range of $a_f$ parameters, for all parameter sets (between $a_f = 0.025$ and $a_f = 0.04$

depending on the parameter set). Decreasing again the $a_f$ parameter produces longer periods. The isolation of a preferential periodicity is not as clear as for the 2.4, 4.8 and $\sim$7 Myr frequencies, and the rise of the dominant frequency of the output signal with the $a_f$ parameter decrease becomes more continuous. However, for each parameter set there is a small step around the fourth multiple of 2.4 Myr, $\sim$ 9.6 Myr (between 9 and 10 Myr, depending on the parameter set). The corresponding $a_f$ interval is much smaller than in the case of the 4.8 and $\sim$7 Myr periods. As the $a_f$ parameter decreases again, the evolution

of the dominant frequencies seems to vary in a non continuous way, with preferred frequencies, but the behaviour is less clear. The presence of preferred period suggests a mechanism of frequency locking and periodicity doubling, which allows to obtain multiple of the 2.4 Myr eccentricity frequency.

As a response to a simple eccentricity forcing, our non linear model is able to produce multi-million year cycles of periods absent of the input forcing (or present with very low amplitude), most probably via a mechanism of period doubling.

However, the shape of the oscillations obtained is quite far away from the data. For each parameter set, there are values of the $a_f$ parameter that produce $\sim$9 Myr cycles in the $\delta^{13}C$. One of them is displayed in Figure 11. For comparison, $\delta^{13}C$ from deep sea records (Zachos et al., 2001; Westerhold, 2020) are also shown. In the modelled $\delta^{13}C$, there are oscillations between high and low $\delta^{13}C$ value, and the shift between the low and high values is quite sudden. This is due to the fact that in our model the $\delta^{13}C$ has two equilibrium values, when the organic matter burial is not forced astronomically. When the

astronomical forcing is added, oscillations occur around one equilibrium value, until the astronomical forcing becomes strong





|  | set 1 | set 2 | set 3 | set 4 | set 5 |
|---|---|---|---|---|---|
| $\frac{a}{a_{lim}}$ | 0.5 | 1.5 | 0.2 | 0.5 | 1.5 |
| $\delta$ | 1.3e-7 | 1.3e-7 | 2e-7 | 2e-7 | 2e-7 |

**Table 1.** Value of the different parameter sets displayed in Figure 10.

enough to push the system towards the second equilibrium. 400 kyr cycles are also noticeable on top of the multi-million year oscillations. However, their strength is underestimated compared to the data.

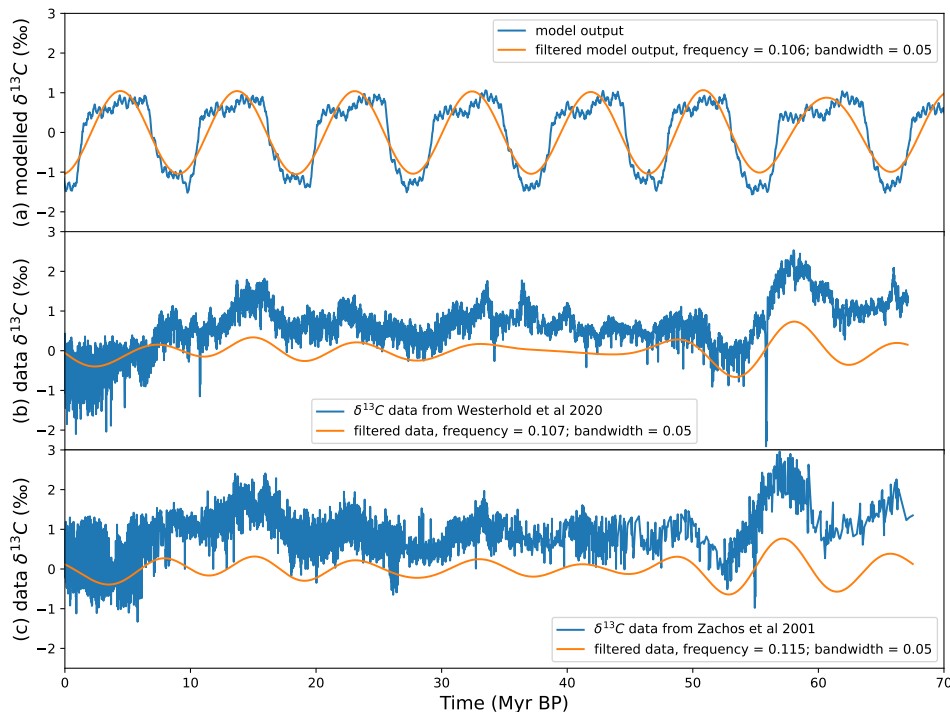

**Figure 11.** Comparison of (a) the modelled $\delta^{13}C$ and (b) deep sea record $\delta^{13}C$ (data are from Westerhold (2020)) in blue and filtered (frequency = 0.107; bandwidth = 0.05) in orange. The modelled $\delta^{13}C$ shown was taken among the model outputs that produces $\sim 9$ Myr cyclicities.

At these timescale, many other processes not taken into account here, such as plate tectonics (Müller and Dutkiewicz, 2018), or other geochemical cycles could play a role and influence the carbon cycle. Our model is simple and excludes many pro-
cesses that could be of importance, and other mechanisms can be considered to produce an internal oscillatory dynamics. In this model, we have linked the carbon and oxygen cycle through the organic matter burial term. The "real" dependency of

the net organic carbon burial B to the surface carbon C and thus climate is probably much more complicated than the shape envisaged here. However, the results obtained are generic, and multiple equilibria and self-sustained oscillations (without any external forcing like eccentricity) could be obtained in the Earth system with a different shape of B corresponding to different

mechanisms. One of the conditions to allow for multiple equilibria to exist is to have a non monotonic dependency of B(C). That B(C) has the same shape as the one assumed in this study is highly unlikely, but the fact that B varies in a non monotonic way with climate and thus the carbon content C is highly probable, due to the variety of processes involved, that might be favored for certain climate and thus lower or higher carbon values. One could also imagine to obtain more complex oscillations involving the sulfur cycle. Oxidation of sulfide minerals produces sulfuric acid, which can react with carbonate minerals and

thus release $CO_2$ (Torres et al., 2014). And sulfide oxidation also affects the oxygen content. Here, we have based the oscillations on the (C,O) elements, but other mechanisms could certainly be considered.

## 4    Conclusions

Multi-million year oscillations are present in the $\delta^{13}C$ records throughout the Cenozoic and part of the Mesozoic. It has been

suggested that these variations could be related to astronomical forcing of organic matter burial fluxes. However, no modelling study has been able to perform such long oscillations through astronomical forcing of organic matter fluxes. Here, we have proposed a mechanism that could explain the presence of multi-million year cycles in the $\delta^{13}C$ record, and their stability over time, as a result of preferential periodicity locking to multiples of astronomical forcing periods. A simple, linear astronomical forcing cannot alone produce multi-million year cycles longer than 2.4 Myr with a strong amplitude, but the presence of

multiple equilibria in our model allows to extract longer periods. Our result show that astronomical forcing, superimposed to internal oscillations of the climate system (like here with carbon and oxygen) is a way to obtain very long term cycles on $\delta^{13}C$, with periodicities that are not directly present in the initial astronomical forcing.

*Code and data availability.* The model code as well as model outputs and figure code is available online : https://doi.org/10.5281/zenodo.7129166.

*Author contributions.* GL and DP designed the study. GL performed the simulations, and wrote the manuscript under the supervision of DP.

*Competing interests.* The authors declare that they have no conflict of interest.

*Acknowledgements.* We acknowledge the use of the LSCE storage and computing facilities and thank ANDRA for their financial support.





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
