# Peer review of "Multi-million year cycles in modelled $\delta^{13}C$ as a response to astronomical forcing of organic matter fluxes."

_Earth System Dynamics, 2022_

## Author Response (AR1)

We thank the reviewers and editor for their helpful comments and suggestions. In the following, the reviewer's or editor's comments are in black, our answer in blue, and the changes made to the manuscript are in green.

G. Leloup and D. Paillard

Answer to RC 1 :

First, as a general comment, we would like to emphasize that our simple model is obviously not designed to be a faithful representation of reality. From a practical point of view, the actual processes involved are far too numerous, they depend on quite local and specific phenomena, and more importantly current knowledge of the long term organic carbon cycle is far too incomplete. We therefore fully agree with both reviewers that in many ways this model is certainly oversimplified. In particular, it is certainly not suited to describe faithfully all the variations in carbon isotopes observed in the geological record.

But our objective is much more modest : we are trying to provide a new framework to explain the persistent long-term (8-9 Myr) oscillations observed over the Cenozoïc and Mesozoïc. The main difficulty is that there is no known external forcing at this particular periodicity. This stands in sharp contrast with the 400 kyr and the 2.4 Myr $\delta^{13}C$ oscillations that can easily be related to the astronomical (eccentricity) forcing. Still, these long-term (8-9 Myr) $\delta^{13}C$ oscillations appear remarkably persistent despite major changes in continental configuration, biological evolution or climate. The suggestion that they might also be astronomically paced is therefore worth examining. Unfortunately, current carbon models do not allow for dynamical behaviors like period doubling or frequency locking : they can generally produce oscillations only at the same frequency as the forcing. If we still wish to explain the observed 8-9 Myr oscillations by some astronomical forcing, we need a model with more varied dynamical behaviors. Our model exemplifies such a possibility.

In a revised version of the manuscript, we would emphasize more on the philosophy of our model and its purpose.

The philosophy of the model has been emphasized at the end of the introduction.

l 73-79 : "Our model is simple and is not designed to be a faithful representation of reality. Rather, we try to produce the simplest model possible that can produce results qualitatively similar to the carbon isotope record, while being compatible with biogeochemistry. This type of approach has been used by Bachan et al (2017) for a different purpose : explain $\delta^{13}C$ excursions during the Mesozoic, having duration of 0.5 to 10 Myr, and declining amplitude over time. Our model is not suited to represent specific excursions in $\delta^{13}C$ , due to particular events of organic matter burial. In this paper we rather focus on the persistent multi-million year cyclicity observed in $\delta^{13}C$ over the last ~200 Myr, over the Cenozoic and latest Mesozoic (Boulila et al, 2012; Martinez and Dera, 2015)" .

Leloup and Paillard present a new model to link astronomical forcing with multi-million-year oscillations in Earth's carbon cycle. I will immediately admit to not being an expert on

astronomical forcing of Earth's surface environment or the mathematical modelling of how different modulations may have influenced surface processes. As such, it is difficult for me to make detailed comments about the modelled approach. However, I do have some general thoughts on the assumptions made by the model.

The authors are very open about the fact that this is a very simple model and that they have been unable to include several processes that may complicate the relationship between astronomical forcing and the carbon cycle. I am the first to acknowledge that any model of geological processes has to make simplifications, and that it is impossible to consider every possible control. However, I do worry that there are some potentially major factors that have not been considered, and whose exclusion from the model makes it potentially unrealistic.

Firstly, and very importantly, whilst the rate of organic-carbon burial is indeed related to global oxygen levels, the reverse is also true. Several studies have highlighted that a large increase in organic-matter deposition will cause surface oxygen levels to rise (see e.g., Lenton and Watson, 2000, Global Biogeochemical Cycles; Berner, 2004, Oxford University Press; but there are many others). It's not clear whether the authors have considered this as a two-way process.

In our study, not only are the organic carbon burial rates dependent on oxygen levels, but oxygen levels also depend on organic matter burial and oxidation. Lines 103 - 105, we state that "On one side, the burial of organic matter is facilitated in low oxygen zones. On the other side, organic matter oxidation reduces the oxygen quantity, while burial of organic matter adds oxygen to the surface system". The influence of organic matter burial on oxygen levels is then reflected in Eq. (5) : $dO/dt = B - Ox$. For positive organic carbon fluxes (net burial higher than net oxidation), the oxygen quantity increases and conversely. The influence of oxygen on organic matter burial is reflected in Eq. (6) : $B(C, O_1) = B(C, O_2) - \delta (O_1 - O_2)$. This dependance to the global oxygen level O is simple in our model (as discussed afterwards) : the organic matter burial decreases linearly with global oxygen contents.

On the subject of oxygen, at the moment the model seems to consider surface oxygen as a single inventory of oceanic and atmospheric oxygen levels, but the reality is that different parts of the marine realm can be very oxygen depleted regardless of overall oxygen levels. This is particularly the case for for small restricted epicontinental basins, and these varied in abundance due to tectonic configuration at various times in Earth's past, and were highly influenced by local processes and sea level changes, both of which are related to astronomical forcing.

Indeed, we fully agree with the reviewer's comment. Higher oxygen levels globally do not necessarily lead to higher oxygen levels in marine parts relevant to organic matter burial. However, it is extremely difficult to account for the numerous important local processes that control oxygen levels and ultimately the burial of organic matter. The simplest possible assumption is therefore to use a global oxygen inventory. But more importantly, our goal is not to have a realistic complex model that represents the oxygen concentration spatially. As explained in the first paragraph, our model is an illustration of the possible role of non-linearities and multiple equilibria to address the question of very long term $\delta^{13}C$ variations, a possibility that, to our knowledge, was never considered before.

The hypothesis of the link between surface oxygen and oxygen at oceanic depth, was emphasized in the revised version of the manuscript. Former l. 132-133 (now l. 149 - 154) were modified to "Organic matter burial is facilitated in locally lower oxygen concentrations. We make the assumption that, at first order, a higher oxygen content globally in the atmosphere leads to higher oxygen contents locally in the ocean, and thus more burial of organic matter in the ocean. In reality, the local oxygen concentrations can differ widely from the global oxygen levels. However, the objective of our model is solely illustrative, and we do not aim at modelling the spatial evolution of oxygen concentrations, and limit ourselves to a single surface oxygen inventory, O. Therefore, in our model, organic matter burial in the ocean decreases for higher oxygen concentrations and inversely."

Also, the authors consider oxidation of non-carbon elements as an important control, but not the potential reduction of these elements, which I find curious. And what about sulfur and phosphorus?

We make no assumption on the sign of our « Ox » term in equation (5) and it therefore implicitly represents the net effect of all processes other that organic burial. These include both the oxidation of non-carbon elements, but also reduction processes. Though the net flux represents on average an « oxidation », it is probably misleading to call it « Ox ». In a revised version of the paper, we will replace « Ox » by « Redox » to avoid misunderstanding.

Ox was replaced by Redox, and former l. 110 (current l.127), we replaced "oxidation of other elements than carbon" by "oxidation and reduction"

I also wonder if the authors have considered the potential role of terrestrial organic-matter burial in their model. Of course, organic carbon burial in the ocean will typically be the more important sink, but there are times in Earth's history when terrestrial burial is thought to have had a massive influence on the global cycle, most famously during the Late Devonian–Carboniferous, but also in the Mesozoic (e.g., Valanginian; see Westermann et al., 2010, EPSL). This is important because the terrestrial sink is likely controlled by very different factors (not directly linked to surface oxygen) than the marine sink.

We agree with the reviewer that terrestrial organic matter certainly plays an important role, in particular at some specific times in the past. But this terrestrial sink is likely to be even more difficult to represent in simple, global terms with an idealized model. Besides, our goal is to investigate the seemingly robust relationship between astronomical forcing and organic matter burial, something which is more likely to originate in the more « stable » oceanic environment. We acknowledge that this model will not be able to represent specific peaks in the $\delta^{13}C$ due to terrestrial organic matter burial variations. This would be emphasized in a revised version of the paper.

The fact that our model will not be able to represent specific peaks in the $\delta^{13}C$ due to terrestrial organic matter burial variations has been emphasized at the end of the introduction. New l. 76 - 77 : Our model is not suited to represent specific excursions in $\delta^{13}C$, due to particular events of organic matter burial.

If it isn't possible to incorporate these factors into the model, then at the very least there needs to be more open consideration of them (as well as other processes which will vary over time). But as things stand, I worry that the list of missing controls is so long at present

that the model cannot really be a strong representation of reality, and that at least some of them need to be included as separate terms regarding the sources and sinks of carbon and oxygen etc.

As explained in the first paragraph, our goal is not to describe all the variations in carbon isotopes observed in the geological record, but to provide a new framework to explain the persistent long-term (8-9 Myr) oscillations observed over the Cenozoïc and Mesozoïc, as a consequence of orbital forcing. We would emphasize on the model objectives in a revised version of the manuscript.

The model's objectives have been emphasized at the end of the introduction in the revised version.

Minor comments:

Line 51: Here 'favour' is written. Elsewhere it is 'favor'. Be consistent.

This will be corrected.

Line 99: A constant fractionation factor of -25 per mil for organic matter is a probably a big assumption given the differences in different organisms, and especially following the rise of C4 plants in the Cenozoic (considering that this paper discusses that time interval).

The value of the fractionation factor could indeed be changed for a lower value, but this would only change the numerical results, not the qualitative oscillations obtained with our model. This remark would be added in a revised version of the manuscript.

Former l 99 -101 (now l. 114 - 117) were modified to :

"We have assumed a constant -5‰ volcanic source ($\delta^{13}V$ = -5‰) and a constant organic matter fractionation. This slightly differs from the equation used in Paillard (2017), however the results remain very similar (for a detailed discussion, see the interactive discussion of the Paillard (2017) paper). The value of the organic matter fractionation factor, $\delta^{13}B$ - $\delta^{13}C$, is set to -25‰. Using another value would change the numerical values of the results, but not their qualitative behavior."

Line 117: How is the carbon cycle forced astronomically? Simply invoking an unnamed link feels rather vague to me.

The assumption that the organic matter fluxes are forced astronomically comes from the persistent observation of 400 kyr cycles in $\delta^{13}C$, and the fact that this frequency is the dominant frequency of eccentricity. Therefore, we chose to force the organic matter flux with eccentricity. However, the 8-9 Myr cycles that are the focus of this paper are not easily explained by a forcing by eccentricity, as the 8-9 Myr frequency is absent from the eccentricity spectra.

Different causal mechanisms have been proposed by authors to explain the link between eccentricity and organic matter fluxes, as explained in the introduction. For instance, Kocken

et al (2019) suggested that marine organic matter burial is enhanced for low eccentricity values, as they could favor annual wet conditions and clay formation, and that the majority of organic carbon is buried in association with clay particles (Hedges and Keil, 1995). Alternatively, Martinez and Dera (2015) suggested that low eccentricity values lead to favorable conditions for persistent anoxia throughout the year, which leads to higher carbon burial in the ocean. In both these cases, lower eccentricity values are associated with higher organic carbon burial, and conversely high eccentricity values are associated with lower organic carbon burial, and the organic matter fluxes are thus "forced astronomically". Our study does not allow us to discriminate between these mechanisms and to say if one is more plausible than the other or if both are at play. We deliberately do not choose a specific physical mechanism, and we rather focus on the implications of having organic matter fluxes that are forced astronomically on the output $\delta^{13}C$ signal, in the case where there are multiple equilibria in the (C,O) system.

Former l. 117 (now 134), we have explicitly added that the astronomical forcing used here is eccentricity : "As in the Paillard (2017) model, the organic matter flux is forced astronomically, with the eccentricity."

l. 142 - 144, we added : "Different mechanisms have been proposed to explain the link between organic matter flux and astronomical forcing, with for instance low eccentricity values being associated with a strong organic matter burial (Martinez and Dera 2015; Kocken et al, 2019). Here, we do not intend to focus on a specific biogeochemical mechanism. We rather focus on the output signal that can be obtained from a simple astronomical forcing. Therefore, we have chosen the easiest possible relationship : a linear variation of the organic matter flux B with the eccentricity."

Lines 132–133: Yes, but this will not be evenly distributed and even when surface oxygen levels rise, there can still be places in the ocean that can be very anoxic.

Indeed, but as explained earlier it is extremely difficult to account for the numerous important local processes that control oxygen levels and ultimately the burial of organic matter and the simplest possible assumption is to use a global oxygen inventory. In a revised version of the manuscript, we will emphasize more on this hypothesis.

l. 132-133 were modified to "Local oxygen concentrations can differ widely from the global oxygen levels. The objective of our model is solely illustrative, and we do not aim at modelling the spatial evolution of oxygen concentrations, and limit ourselves to a single surface oxygen inventory, O. We make the assumption that, at first order, a higher oxygen content globally in the atmosphere leads to higher oxygen contents locally in the ocean"

Line 154–155: This will then cool the climate and reduce organic-matter oxidation, raising surface oxygen levels, both of which will act to mitigate the organic-carbon burial.

This would be true in a model with a single equilibria, if B increased monotonically with C. However, in our model B does not only represent the marine organic matter burial, it is the difference between organic matter burial $B^+$ (that includes terrestrial burial, and oceanic burial of organic matter of both terrestrial and marine origin) and oxidation ($B^-$), $B = B^+ - B^-$.

In the study, we do not make particular assumptions on the evolution of terrestrial burial with carbon (climate) and oxygen contents. We assume that both organic matter oxidation ($B^-$) and organic matter burial in the ocean (and thus $B^+$) increase with increasing C.

If the oceanic carbon burial $B^+$ increases with C, but that $B^-$ increases more sharply with C (which is the case for intermediate carbon values, $C_1 < C < C_2$, in our model and depicted in Figure RC1), warmer temperatures (higher C) do not lead to an increased organic matter flux B, but a decreased one.

[Figure]

Fig RC1 : Schematic representation of the evolution of organic matter burial ($B^+$), organic matter oxidation ($B^-$) and organic matter flux ($B = B^+ - B^-$) with surface carbon.

In the revised version, we have clarified the definition of the organic matter flux (former l. 89 - 90, now l.95-105)

"The organic matter flux B represents all organic carbon fluxes to and from the surface system. It is composed of two opposite contributions, $B = B^+ - B^-$, where $B^+$ represents organic carbon burial and $B^-$, represents organic matter oxidation. Thus the organic matter flux B is positive when there is a net burial and negative when there is a net oxidation of organic matter.

Organic matter burial, $B^+$, is composed of terrestrial burial, as well as oceanic burial of organic matter of both terrestrial and marine origin. For instance, eroded terrestrial organic matter from plants is delivered to rivers (Meybeck, 1982; Ludwig et al, 1996). If a part of this biospheric organic carbon is buried into sediments without being degraded, this corresponds to a decrease of the surface carbon content. It has been estimated that the current burial flux of organic carbon eroded from land into oceanic sediments is around 40-80 MtC/yr (Hilton and West, 2020). Organic matter oxidation, $B^-$ can come from the exhumation of sedimentary rocks and the oxidation of petrogenic organic carbon, leading to $CO_2$ release in the atmosphere (Hilton et al, 2014). The carbon flux released to the atmosphere through

petrogenic organic carbon oxidation has been estimated to be between 40 and 100 MtC/yr (Hilton and West, 2020)."

Also, the dependency of B+ and B- to C and O resulting in a dependency of B to C and O has been clarified (l. 149 - 159 for the dependency of B to O, and l. 160 - 185 for the dependency of B to C)

Lines 162–168: What are 'lower', 'intermediate', and 'higher' carbon values defined as? What range?

In a revised version of the paper, the values of $C_1$ and $C_2$, ie the ranges for "lower" ($C < C_1$), "intermediate" ($C_1 < C < C_2$) and "high" carbon values ($C > C_2$) will be indicated more clearly in Section 2.2, by indicating the numerical values. For now, it is written $C_1 = C_{eq1ref} + (1/3)$ ($C_{eq2ref} - C_{eq1ref}$) and $C_2 = C_{eq1ref} + (2/3)$ ($C_{eq2re}f - C_{eq1ref}$). The corresponding numerical value of $C_1 = $ 44 333 PgC and $C_2 = $ 45 667 PgC will be added.  Also, the reader will be referred to section 2.2 at the first mention of low, intermediate and high carbon values, for more clarity. However, we emphasize that these values are model parameters that could be changed. This would change the numerical value of the oscillations obtained, but does not change the main result of the paper : being able to obtain longer oscillations than present in the input forcing, due to the presence of multiple equilibria in the carbon cycle.

The numerical values of $C_1 = $ 44 333 PgC and $C_2 = $ 45 667 PgC have been added in section 2.2.

Line 175: '…we place ourselves here in one of the simplest case possible.' is rather casual language for me.

This formulation will be modified in a revised version of the manuscript.

This has been replaced by "we have chosen one of the simplest case possible"

Line 201: I assume that the organic-matter burial being referred to here is oceanic. What about terrestrial organic-carbon burial?

In our model, the terrestrial organic matter burial does not vary with oxygen, nor does the organic matter oxidation. However, the organic matter burial in the ocean is influenced by the oxygen content O (in a linear way in our model), so the total organic matter flux (sum of marine and terrestrial organic matter burial minus oxidation) varies with oxygen. This should be clarified in a revised version of the manuscript, when introducing the dependence of B to O (starting l. 132).

The fact that in our model, only organic matter burial in the ocean varies with oxygen (decreases with increasing oxygen), and that the other fluxes do not vary with oxygen, resulting in a decrease of the total organic matter flux B with oxygen, has been clarified :

 l. 153-156 "Therefore, in our model, organic matter burial in the ocean decreases for higher oxygen concentrations and inversely. We have assumed that other organic matter fluxes do not vary with oxygen, and thus the decrease of marine organic matter burial with oxygen leads to a decrease of the net organic matter flux B with oxygen."

Line 258: What about reduction of other elements?

In a revised version of the paper, we will replace « Ox » by « Redox » to avoid misunderstanding.

Ox was replaced by Redox, and former l. 110 (current l.127) we replaced "oxidation of other elements than carbon" by "oxidation and reduction"

Line 422: I'm not sure a mechanism is being proposed per say. It's been assumed that astronomical forcing influences carbon supply vs burial and oxygen levels, and that long-term cycles can be reproduced for a certain set of parameters. But this is all very theoretical still and there isn't a cause-and-effect link proposed for how the astronomical forcing is influenced these carbon and oxygen sources and sinks. For me, that would be the mechanism.

In this sentence, mechanism does not refer to a physical mechanism linking organic matter fluxes to astronomical forcing. Rather, it is meant as a "dynamic" mechanism, that allows to obtain oscillations with longer periods than the input forcing : in our case, the presence of multiple equilibria in the (C,O) system, that can lead to longer oscillations when an astronomical forcing is added. The sentence could be clarified by using the term "framework" instead of "mechanism", and we could emphasize the link with the explanation in l. 425.

Former l. 422 (now 443) reads : "Here, we have proposed a mathematical mechanism, compatible with biogeochemistry, that could explain the presence of multi-million year cycles in the $\delta^{13}$C record, and their stability over time, as a result of preferential phase locking to multiples of the 2.4 Myr eccentricity period."

Refs :

Hedges and Keil (1995), Sedimentary organic matter preservation: an assessment and speculative synthesis, Marine Chemistry, 49, 81-115

Kocken et al (2019), The 405 kyr and 2.4 Myr eccentricity components in Cenozoic carbon isotope records, Climate of the Past, 15, 91-104

Martinez and Dera (2015), Orbital pacing of carbon fluxes by a ~9 Myr eccentricity cycle during the Mesozoic, Proceedings of the National Academy of Sciences, 112, 12604 - 12609

Answer to RC 2 :

First, as a general comment, we would like to emphasize that our simple model is obviously not designed to be a faithful representation of reality. From a practical point of view, the actual processes involved are far too numerous, they depend on quite local and specific phenomena, and more importantly current knowledge of the long term organic carbon cycle is far too incomplete. We therefore fully agree with both reviewers that in many ways this model is certainly oversimplified. In particular, it is certainly not suited to describe faithfully all the variations in carbon isotopes observed in the geological record.

But our objective is much more modest : we are trying to provide a new framework to explain the persistent long-term (8-9 Myr) oscillations observed over the Cenozoïc and Mesozoïc. The main difficulty is that there is no known external forcing at this particular periodicity. This stands in sharp contrast with the 400 kyr and the 2.4 Myr 13C oscillations that can easily be related to the astronomical (eccentricity) forcing. Still, these long-term (8-9 Myr) 13C oscillations appear remarkably persistent despite major changes in continental configuration, biological evolution or climate. The suggestion that they might also be astronomically paced is therefore worth examining. Unfortunately, current carbon models do not allow for dynamical behaviors like period doubling or frequency locking : they can generally produce oscillations only at the same frequency as the forcing. If we still wish to explain the observed 8-9 Myr oscillations by some astronomical forcing, we need a model with more varied dynamical behaviors. Our model exemplifies such a possibility.

In a revised version of the manuscript, we would emphasize more on the philosophy of our model and its purpose.

**Peer review of „Multi-million year cycles in modelled d13C as a response to astronomical forcing of organic matter fluxes".**

In this paper, the authors built a simplified numerical representation of the carbon cycle, assuming a mass balance without carbon reservoirs (and hence no lag-times there), unlimited nutrients (otherwise organic burial B would also depend on weathering W), and with constant $[Ca^{2+}]$ concentration in the ocean.

>> assuming a mass balance without carbon reservoirs (and hence no lag-times there)

In the model, we have one global carbon reservoir and one global oxygen reservoir, with the associated time lags : in particular, this is what enables a self-sustained oscillation regime.

>> unlimited nutrients (otherwise organic burial B would also depend on weathering W),

We have no « explicit » weathering, but we are assuming that climate warms when the global carbon content C increases, therefore the traditional Walker feedback through an increase in the carbonate precipitation. We actually have the same feedback on the organic carbon via the relationship between burial B and carbon C : when climate warms, this induces several processes (hence a non-monotonous relation) among which the increase in weathering and nutrient supply. This is further detailed in the dedicated comment on the decoupling between B and W.

>>with constant $[Ca^{2+}]$ concentration in the ocean

We indeed assume a constant $[Ca^{2+}]$ concentration in the ocean. This is a limitation of our model. In particular, including calcium variations could change (and complexify) the relationship between global carbon content C and atmospheric $CO_2$, and therefore the climate forcings. This will be explained more clearly in the revised manuscript.

This is now mentioned l. 425-427 : "For instance, we have assumed a constant $Ca^{2+}$ concentration in the ocean. This is a limitation of our model. In particular, including calcium variations could change and complexify the relationship between global carbon content C and atmospheric $CO_2$."

Without applying any forcing, their model evolves into steady-state equilibrium when the oxidation of other elements than organic carbon (Ox) increases steeply with oxygen content (O). When the Ox term increases less steeply with O, the model produces oscillations in d13C without any astronomical forcing. Finally, the authors add an eccentricity forcing to the burial of organic carbon and they observe that the resulting d13C is oscillating with preferential periodicities of 2.4, 4.8 and 7.2 Myr. The authors thus built a model that is prone to oscillate at multi-million-year timescales between multiple equilibria, and by adding the forcing they are making sure that the model resides around one equilibrium value until the astronomical forcing becomes strong enough to push the system towards the second equilibrium. Finally, the authors compare their model results to the Westerhold et al. (2020) benthic d13C compilation and point out to the reader that the multi-million-year oscillations in this record could be the result of self-sustained oscillations in the Earth system.

We would like to clarify that in our study, we do not suggest that the multi-million year oscillations observed in the $\delta^{13}$C record are the direct result of self sustained / internal oscillations in the Earth system.

Rather, we suggest that the addition of astronomical forcing to a system with multiple equilibria can produce oscillations in the $\delta^{13}$C with periodicities that are different from the astronomical periodicities.

In our case, the carbon-oxygen system without astronomical forcing can produce self-sustained oscillations under certain parameter values (when the Ox term increases less steeply with O, ie if $a < a_{lim}$) but does not necessarily (no oscillations are obtained if $a > a_{lim}$). However, in both cases (self sustained oscillations or not), the addition of astronomical forcing to the system changes its behaviour and leads to oscillations in the $\delta^{13}$C that can have different frequencies than the astronomical one, and that have a different period than the self sustained oscillations in the case where they exist.

The fact that we can obtain oscillations with astronomical forcing even when there are no oscillations in the unforced system has been clarified, l. 391-393 : "In case B, it is worth emphasizing that even if there are no self sustained oscillations without astronomical forcing, the addition of the astronomical forcing of the organic matter flux also leads to multi-million year cycles in $\delta^{13}$C."

**Major concern.**

This is a nice "back-of-the-envelope" carbon cycle exercise, but I do not see the immediate merit in this paper. The authors set the model variables such that it is prone to produce multi-million-year cycles. They force it with an eccentricity cycle (including the 2.4 Myr component) and come back home with a simulated d13C signal that emphasizes these same 2.4 Myr cycles, as well as multiples of that cycle. I would be interested to read why the authors believe their approach provides additional insights into the behavior of the carbon cycle in addition to other previous attempts to simulate the global carbon cycle.

In contrast to many previous attempts, we simply do not attempt to « simulate » the global carbon cycle. We try to provide a possible theory that could explain the occurrence of persistent long-term oscillations, at periodicities roughly a multiple of the forcing. To our knowledge, this has never been attempted in carbon cycle studies before.

We want to emphasize that our major modeling assumption (multiple equilibria) is rather natural though unconventional. Indeed, most carbon cycle models are built on the premises that they should exhibit one equilibrium (and if possible an equilibrium that resembles the current state when submitted to present-day forcing). But net organic matter burial depends upon climate in numerous fashion that acts either ways, with warming favoring burial or favoring old carbon remineralization. Overall, it is unlikely that the relationship between organic matter burial and climate is always monotonous. As shown in our manuscript, assuming such a non-monotonous relationship leads quite naturally to multiple equilibria in our simple carbon-oxygen model, something which may explain some features of past carbon cycle changes.

We force our model with an eccentricity cycle, that includes a 2.4 Myr component, and produce $\delta^{13}C$ cycles of 2.4 Myr, and preferentially multiples of 2.4 Myr. However, this finding already differs from previous studies, such as Paillard (2017) or Kocken et al (2019). In these studies, the organic matter burial was also forced with an eccentricity cycle, but the obtained $\delta^{13}C$ cycles did not contain periodicities longer than 2.4 Myr. Being able to produce periodicities longer than 2.4 Myr with an eccentricity forcing is not a trivial result. It is possible in our case, because the model is non linear and contains multiple equilibria. In the models of Paillard (2017) and Kocken (2019), where the formulations are mostly linear it is not possible to produce $\delta^{13}C$ oscillations with periods longer than 2.4 Myr by forcing solely with the eccentricity.

I am especially thinking about Bachan et al. (2017), who reports on carbon cycle stabilization pathways in response to a sinusoidal forcing.

We thank the reviewer for the suggestion of the Bachan et al (2017) article. Our study shares a similar philosophy with the one of Bachan (2017). Indeed, Bachan (2017) states that "*Many sophisticated models have been put forth to interpret geochemical record and simulate global biocheochemical dynamics (BLAG, Berner and others 1983, Copse, Bergman and others 2004, MAGic, Arvidson and others 2006). The goal here is not to replicate these models. Rather, our goal is to produce the simplest possible model that still bears a semblance of the physical system being modeled, and can produce results that are qualitatively similar to the carbon isotope record*".

In this study, we also look for the simplest possible model that can explain observed features of the $\delta^{13}C$ record. However, our model and the one of Bachan (2017) have different goals

and make different assumptions. Bachan (2017) focuses on $\delta^{13}$C excursions, having durations of 0.5 to 10 Myr, and declining amplitude over time, taking place mostly in the Paleozoic and the earliest part of the Mesozoic. There are no marked excursions in the $\delta^{13}$C record over the last 200 Myr.

In our study, we focus on multi-million year cycles in the $\delta^{13}$C over the last ~200 Myr (Cenozoic and latest Mesozoic), as it is the period on which oscillations of 8-9 Myr in the $\delta^{13}$C have been observed (Boulila et al (2012) for the Cenozoic, and Martinez and Dera (2015) for the period from 130 Myr BP to 200 Myr BP).

The observed $\delta^{13}$C oscillations are of lower amplitude than the $\delta^{13}$C excursions. The amplitude of the oscillations is around 2‰, while the positive $\delta^{13}$C excursions in the Earliest Phanerozoic have amplitudes of 5-10‰.

In the study of Bachan (2017), there are no multiple equilibria. The system is linear, forced with a sinusoidal forcing. In the Bachan (2017) study, a resonance behaviour is observed. Larger amplitudes of $\delta^{13}$C oscillations are obtained for larger amplitudes of the sinusoidal forcing, and this is especially true for input frequencies close to the resonant frequency, where the $\delta^{13}$C oscillations amplitude changes due to amplitude variation of the input forcing are amplified. However, this differs from our study, as the output $\delta^{13}$C signal oscillations obtained in Bachan (2017) always have the same frequency as the sinusoidal input forcing. In our case, changing the amplitude of the input forcing (the $a_f$ parameter) does not change much the amplitude of the $\delta^{13}$C oscillations, But the novelty of our study is that by changing the amplitude of the input forcing (by modifying the $a_f$ parameters that controls the strength of the astronomical forcing, the eccentricity, in our case) we produce $\delta^{13}$C oscillations that have a dominant frequency that is not present in the input forcing (the eccentricity in our case). Bachan (2017) suggests that linear resonance might be an important concept to explain some high amplitude 13C excursions in the Paleozoïc. We are suggesting that period-doubling and multiple equilibria might be an important concept to explain the persistent long-term 13C oscillations observed at least since the Mesozoïc up to now.

In a new version of the manuscript, we would emphasize on the fact that our interest lies in oscillations in $\delta^{13}$C over the last ~200 Myr, period for which 8-9 Myr oscillations of the $\delta^{13}$C have been reported [Boulila et al (2012), Martinez and Dera (2015)].

The objective of our model and its difference to the one of Bachan et al (2017) has been emphasized at the end of the introduction.

l 73-79 : "Our model is simple and is not designed to be a faithful representation of reality. Rather, we try to produce the simplest model possible that can produce results qualitatively similar to the carbon isotope record, while being compatible with biogeochemistry. This type of approach has been used by Bachan et al (2017) for a different purpose : explain $\delta^{13}$C excursions during the Mesozoic, having duration of 0.5 to 10 Myr, and declining amplitude over time. Our model is not suited to represent specific excursions in $\delta^{13}$C , due to particular events of organic matter burial. In this paper we rather focus on the persistent multi-million year cyclicity observed in $\delta^{13}$C over the last ~200 Myr, over the Cenozoic and latest Mesozoic (Boulila et al, 2012; Martinez and Dera, 2015)" .

I also feel that some simplifications in the model need to be more clearly justified. It seems contra-intuitive to de-couple silicate weathering from the organic carbon flux (B does not depend on W). The ocean cannot recycle the same nutrients ad infinitum. You have to introduce new nutrients to compensate for the ones lost to mineralization and burial. Those nutrients come from terrestrial weathering.

Indeed, weathering and nutrients availability influence primary productivity and have thus the potential to impact marine organic matter burial. A lack of nutrients would lead to a lower primary productivity in the ocean. If the  preservation efficiency of marine organic carbon (the ratio of marine organic carbon buried to the marine organic initially produced - the organic carbon primary productivity) remains constant, a lower primary productivity would lead to a lower burial. However, as the preservation efficiency of marine organic matter is very low, only around 0.2 - 1.3% (Burdige 2007, Kandasamy and Nagender Nath 2016), small changes of the organic carbon preservation efficiency can also highly influence the organic matter burial. Thus, the influence of weathering on marine organic carbon burial is not so straightforward, as a decreased marine primary productivity does not necessarily lead to changes in organic matter burial, if there are changes in organic matter burial preservation efficiency due to other environmental factors.

Also, our organic carbon flux term B is a sum of organic matter burial (B+) and oxidation (B-). The organic matter burial can take place on land, or on the ocean. The organic matter buried in the ocean can be of both terrestrial or marine origin. The organic matter of terrestrial origin (approximately one third of organic matter buried in the oceans at present, Burdige 2005) is not influenced by the nutrient changes in the ocean. The organic matter oxidation (B-) does not depend on the nutrient availability in the ocean.

Therefore, there is not a direct dependence of the organic matter flux term (B) to the nutrients in the ocean. Rather in this study, we chose to look at the dependance of organic matter fluxes to climate (through the surface carbon content C), and oxygen O. And climate can influence weathering and thus nutrient availability and primary productivity in the ocean.

In our model, larger carbon values and hotter, wetter climate lead to more marine organic carbon burial for different reasons. First, for warmer temperatures the solubility of oxygen on surface water is decreased (Bopp et al, 2002) and ocean stratification is increased, leading to expansion of oxygen minimum zones (Stramma et al 2008). This increases organic matter preservation. Second, warmer and wetter climate can increase weathering, and thus the delivery of nutrients to the ocean. The consequent increase in primary productivity and oxygen consumption can lead to regional deoxygenation, and thus enhance organic matter preservation (Baroni et al 2020).

However, the global organic matter flux B, that is the difference of the total organic burial B+ and organic matter oxidation B-, does not increase in a monotonous way with warmer climate as marine organic matter burial is not the only process varying with climate.

In our study, we have made the assumption that oxidation of petrogenic organic carbon (B-) also increases with warmer temperatures (larger C contents). Thus, the evolution of organic

matter fluxes with climate depends on both the relative dependence to C of burial and oxidation.

In our study, we make the assumption that for low carbon values, and thus colder climate, the organic matter flux does not vary much with climate. For intermediate carbon C values, we make the assumption that the increase in organic matter oxidation ($B^-$) with temperature and C is steeper than the increase in organic matter burial ($B^+$), which leads to a lower organic matter flux (B) with increasing C. On the contrary, we make the assumption that for higher carbon values, the increase in organic matter burial ($B^+$) is steeper than the increase in organic carbon oxidation ($B^-$), leading to an increase of organic matter flux (B) with increasing C. This is schematized in Fig. RC2.

[Figure]

Fig RC2 : Schematic representation of the evolution of organic matter burial ($B^+$), organic matter oxidation ($B^-$) and organic matter flux ($B = B^+ - B^-$) with surface carbon.

In a revised version of the manuscript, the description of the organic matter flux term B should be clarified. We will emphasize on B being the difference between organic matter burial, $B^+$, (that includes terrestrial burial, and oceanic burial of organic matter of both terrestrial and marine origin) and organic matter oxidation, $B^-$. We do not make particular assumptions on the evolution of terrestrial burial with carbon (climate) and oxygen contents. We assume that both organic matter oxidation ($B^-$) and organic matter burial in the ocean (and thus $B^+$) increase with increasing C. However, if $B^+$ and $B^-$ have different slopes of increase with C, this leads to a non monotonous evolution of $B = B^+ - B^-$ with C. The exact shape of the evolution of B(C) does not impact the results, as long as it is non monotonous, it can lead to multiple equilibria in the carbon cycle (multiple crossing of the red and green curves of Figure 2). We assume that the marine organic matter burial and thus $B^+$ decreases with increasing oxygen levels, resulting in a decrease of B with increasing O.

In the revised version, we have clarified the definition of the organic matter flux (former l. 89 - 90, now l.95-105)

"The organic matter flux B represents all organic carbon fluxes to and from the surface system. It is composed of two opposite contributions, $B = B^+ - B^-$, where $B^+$ represents organic carbon burial and $B^-$, represents organic matter oxidation. Thus the organic matter flux B is positive when there is a net burial and negative when there is a net oxidation of organic matter.

Organic matter burial, $B^+$, is composed of terrestrial burial, as well as oceanic burial of organic matter of both terrestrial and marine origin. For instance, eroded terrestrial organic matter from plants is delivered to rivers (Meybeck, 1982; Ludwig et al, 1996). If a part of this biospheric organic carbon is buried into sediments without being degraded, this corresponds to a decrease of the surface carbon content. It has been estimated that the current burial flux of organic carbon eroded from land into oceanic sediments is around 40-80 MtC/yr (Hilton and West, 2020). Organic matter oxidation, $B^-$ can come from the exhumation of sedimentary rocks and the oxidation of petrogenic organic carbon, leading to $CO_2$ release in the atmosphere (Hilton et al, 2014). The carbon flux released to the atmosphere through petrogenic organic carbon oxidation has been estimated to be between 40 and 100 MtC/yr (Hilton and West, 2020)."

Also, the dependency of B+ and B- to C and O resulting in a dependency of B to C and O has been clarified (l. 149 - 159 for the dependency of B to O, and l. 160 - 185 for the dependency of B to C)

l. 149 - 159 : "Organic matter burial is facilitated in locally lower oxygen concentrations. We make the assumption that, at first order, a higher oxygen content globally in the atmosphere leads to higher oxygen contents locally in the ocean, and thus more burial of organic matter in the ocean. In reality, the local oxygen concentrations can differ widely from the global oxygen levels. However, the objective of our model is solely illustrative, and we do not aim at modelling the spatial evolution of oxygen concentrations, and limit ourselves to a single surface oxygen inventory, O. Therefore, in our model, organic matter burial in the ocean decreases for higher oxygen concentrations and inversely. We have assumed that other organic matter fluxes do not vary with oxygen, and thus the decrease of marine organic matter burial with oxygen leads to a decrease of the net organic matter flux B with oxygen. Here, we have assumed this relationship to be linear."

l. 169 - 185 : "Here, we also suggest that the organic carbon flux depends on the surface carbon quantity C. Indeed, climate can influence organic matter burial and oxidation, and as a first approximation, larger carbon values C in the surface system correspond to warmer, wetter climates.

On one side, higher temperatures and stronger runoff increase erosion and transfer of biospheric organic carbon (Hilton, 2017; Smith et al., 2013). In addition, higher temperatures increase ocean stratification and decrease the solubility of oxygen in surface waters (Bopp et al, 2002), leading to expansion of oxygen minimum zones (Stramma et al., 2008, 2010). This decreases organic matter oxidation and favors its burial into oceanic sediments (Jessen et al, 2017). Other climatic related factors, have been suggested to limit dissolved oxygen in the ocean, such as increased phosphorus inputs (Baroni et al., 2020; Niemeyer et al., 2017). These inputs are expected to increase for warmer and wetter climate, that increases

weathering, leading to regional deoxygenation and organic carbon burial (Baroni et al., 2019). Organic matter burial, B+, is therefore expected to increase with increasing C.

On the other hand, it has also been suggested that the oxidation of petrogenic organic carbon could be linked to climate as petrogenic organic carbon oxidation could be locally limited by temperature, $O_2$ contents, and microbial activity (Chang and Berner, 1999; Bolton et al., 2006; Hemingway et al., 2018; Petsch et al., 2005). . Higher temperature could lead to stronger petrogenic carbon oxidation, and thus organic matter oxidation B- could also increase with increasing C.

The evolution of the total organic matter flux, B = B+ - B-, with C can thus be non trivial as the opposite contributions of B+ and B- can both increase with C. Here, we have made the assumption that for intermediate carbon value, and thus intermediate temperatures, the increase in oxidation of petrogenic organic carbon with temperature is dominant, leading to a stronger increase of B- with C than B+. This results in a decrease of B with C. We make the assumption that for higher temperatures, the increase in export of biospheric carbon and increased burial with temperature dominates. In that case, there is a stronger increase of B+ with C than B-, and this results in an increase of B with C.

Therefore, in our model, for low carbon values (C <C1) and thus colder climates, the organic carbon flux B does not depend on C. Then, for intermediate carbon values (C1 < C < C2), the organic carbon flux decreases with increasing temperatures, and thus increasing carbon C. Finally, for higher carbon values (C > C2) the organic carbon flux increases with increasing temperature (and thus, carbon C). For the sake of simplicity, we have made the assumption of linear variations."

 Moreover, that weathering also modulates the availability of alkalinity, which balances out the atmospheric CO2, and allows for calcification. One ends up with a triangle of calcification (take alkalinity and nutrients, releases CO2), weathering (take CO2, releases alkalinity and nutrients) and organic matter burial (take CO2 and nutrients). But these three are not in phase with each other, which in itself already results in an oscillatory pattern.

Indeed, but these time scales are to short to account for a ~$10^7$ year oscillatory behavior, since carbon or phosphorus have residence times ~$10^5$ years, about 2 order of magnitude more rapid than our phenomenon. This is why the oxygen cycle might play a role in our case. Alternatively, this could be due to other multi million year process.

Bachan, Aviv, et al. "A model for the decrease in amplitude of carbon isotope excursions across the Phanerozoic." American Journal of Science 317.6 (2017): 641-676.

**Minor concern.**

The y-axes in Figures 2 and 3 are incorrectly labeled.

- In Figure 2, the y-axis represents B, not dC/dt. My suggestion would be that the authors hatch the area in-between the organic and inorganic terms and label them with dC/dt>0 when the inorganic term is larger than the organic term, and vice versa.
- In Figure 3, the y-axis represents B for the green curve and Ox for the blue curve. Not dO/dt. Again, here the authors could hatch

We thank the reviewer for this suggestion that we will follow in the revised version.

The y axis of the figures have been modified, and we have more clearly indicated the areas with dC/dt >0 or dC/dt <0 and similarly for oxygen.

References :
Baroni et al (2020), Enhanced Organic Carbon Burial in Sediments of Oxygen Minimum Zones Upon Ocean Deoxygenation, Frontiers in Marine Science, 6
Bopp et al (2002), Climate-induced oceanic oxygen fluxes: Implications for the contemporary carbon budget, Global Biogeochemical Cycles, 16, 6-1-6-13
Boulila et al (2012), A ~9 Myr cycle in Cenozoic $\delta^{13}C$ record and long-term orbital eccentricity modulation: is there a link ? Earth and Planetary Science Letters, 317-318, 273, 281
Burdige (2005), Burial of terrestrial organic matter in marine sediments: A re-assessment, Global Biogeochemical Cycles, 19, 4
Burdige (2007), Preservation of organic matter in marine sediments: controls, mechanisms, and an imbalance in sediment organic carbon budgets?, Chemical reviews, 107, 467-485
Kandasamy and Nagender Nath (2016), Perspectives on the terrestrial organic matter transport and burial along the land-deep sea continuum: caveats in our understanding of biogeochemical processes and future needs, Frontiers in Marine Science, 3, 259
Martinez and Dera (2015), Orbital pacing of carbon fluxes by a ~9 Myr eccentricity cycle during the Mesozoic, Proceedings of the National Academy of Sciences, 112, 12604 - 12609
Stramma et al (2008), Expanding Oxygen-Minimum Zones in the Tropical Oceans, Science, 320, 655-658

**Answer to the editor's comment :**

You have proactively responded to the referees comments and I would like to further encourage you to acknowledge the "simplest model" approach, and in this spirit contrast your model from Bachan's one. Cleary, you have put the basic ingredients of a Fitzhugh Nugomo oscillator, with a bi-stable nullcline for B(C), and dO/dt=B-Ox, which is then phase-locked to the orbital forcing, as well known. In that sense, I sympathises with the reservations of reviewer #2: you want an oscillation, and you make up one, and you need 13 parameters at that. I believe it is very important to be as clear as possible with the implications of this exercise: have you identified a mathematical mechanism (the phase-locked oscillation, in contrast to nonlinear resonance), or a biogeochemical one (the burial dependency)?

The simplest model approach and the differences to Bachan et al (2017) have been emphasized at the end of the introduction.

l 73-79 : "Our model is simple and is not designed to be a faithful representation of reality. Rather, we try to produce the simplest model possible that can produce results qualitatively similar to the carbon isotope record, while being compatible with biogeochemistry. This type of approach has been used by Bachan et al (2017) for a different purpose : explain $\delta^{13}$C excursions during the Mesozoic, having duration of 0.5 to 10 Myr, and declining amplitude over time. Our model is not suited to represent specific excursions in $\delta^{13}$C , due to particular events of organic matter burial. In this paper we rather focus on the persistent multi-million year cyclicity observed in $\delta^{13}$C over the last ~200 Myr, over the Cenozoic and latest Mesozoic (Boulila et al, 2012; Martinez and Dera, 2015)" .

We have emphasized on the fact that we have not identified a biogeochemical mechanism, but a mathematical one, that is compatible with biochemistry. This has been emphasized in the conclusion, l. 443 - 446 :

"Here, we have proposed a mathematical mechanism, compatible with biogeochemistry, that could explain the presence of multi-million year cycles in the $\delta^{13}$C record, and their stability over time, as a result of preferential phase locking to multiples of the 2.4 Myr eccentricity period."

In passing, I very much suspect that the 'period doubling' is not the right concept here. Phase locking (p:q, see Pikovsky) is. Period doubling is perhaps more adequate for unforced regimes (e.g., logistic population dynamics).

We have replaced period doubling py phase locking.

Avoid 'warmer' temperatures. Higher temperatures.

This has been corrected.

current line 103: define B+ and B- explicitly.

In the revised version, we have clarified the definition of the organic matter flux (former l. 89 - 90, now l.95-105)